

# A symmetry principle for gauge theories with fractons

Yuji Hirono[1,2,3★], Minyoung You[1†], Stephen Angus[1‡] and Gil Young Cho[1,2,4∘]

**1** Asia Pacific Center for Theoretical Physics, Pohang, Gyeongbuk, 37673, Korea
**2** Department of Physics, Pohang University of Science
and Technology, Pohang, Gyeongbuk, 37673, Korea
**3** Department of Physics, Kyoto University, Kyoto 606-8502, Japan
**4** Center for Artificial Low Dimensional Electronic Systems,
Institute for Basic Science (IBS), Pohang 37673, Korea

★ yuji.hirono@gmail.com , † minyoung.you@apctp.org ,
‡ stephen.angus@apctp.org , ∘ gilyoungcho@postech.ac.kr

## Abstract

Fractonic phases are new phases of matter that host excitations with restricted mobility. We show that a certain class of gapless fractonic phases are realized as a result of spontaneous breaking of continuous higher-form symmetries whose conserved charges do not commute with spatial translations. We refer to such symmetries as nonuniform higher-form symmetries. These symmetries fall within the standard definition of higher-form symmetries in quantum field theory, and the corresponding symmetry generators are topological. Worldlines of particles are regarded as the charged objects of 1-form symmetries, and mobility restrictions can be implemented by introducing additional 1-form symmetries whose generators do not commute with spatial translations. These features are realized by effective field theories associated with spontaneously broken nonuniform 1-form symmetries. At low energies, the theories reduce to known higher-rank gauge theories such as scalar/vector charge gauge theories, and the gapless excitations in these theories are interpreted as Nambu–Goldstone modes for higher-form symmetries. Due to the nonuniformity of the symmetry, some of the modes acquire a gap, which is the higher-form analogue of the inverse Higgs mechanism of spacetime symmetries. The gauge theories have emergent nonuniform magnetic symmetries, and some of the magnetic monopoles become fractonic. We identify the 't Hooft anomalies of the nonuniform higher-form symmetries and the corresponding bulk symmetry-protected topological phases. By this method, the mobility restrictions are fully determined by the choice of the commutation relations of charges with translations. This approach allows us to view existing (gapless) fracton models such as the scalar/vector charge gauge theories and their variants from a unified perspective and enables us to engineer theories with desired mobility restrictions.

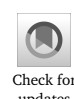
# 1 Introduction

Fractons are excitations with mobility restrictions, and phases with fractons constitute a new class of quantum phases of matter [1, 2]. Gapless fractonic phases have been described by higher-rank gauge theories [3–13], including the scalar/vector charge gauge theories. Higher-rank gauge fields appear as a result of a gauging of the multipole algebra [14–16] (see also discussions on the relation between multipoles and fracton phases in Refs. [17, 18]). As a physical realization of fracton phases, the elasticity theory in two dimensions has been shown to be dual to a symmetric tensor gauge theory, and disclinations in solids are fractonic [19]. This type of duality has been extended to other systems such as supersolids [20] and vortex crystals [21].

In this paper, we show that gapless fractonic phases are realized as a result of spontaneous symmetry breaking (SSB) of continuous higher-form symmetries [22] whose conserved charges do not commute with spatial translations. Such symmetries are a higher-form analogue of spacetime symmetries, and we will refer to them as *nonuniform higher-form symmetries*. As a starting point, we interpret the existence of a fracton as a restriction on the possible configurations of the worldlines of particles. Since they are lines, it is natural to consider 1-form symmetries whose charged objects are the worldlines themselves. Mobility restrictions can be implemented by introducing a continuous nonuniform symmetry whose charge does not commute with translations, $P_i$. For example, if we wish to implement mobility restrictions on the worldline of a particle charged under $Q$, we can make the particle immobile in the $i$-th direction by introducing a symmetry generated by $Q'$ such that $[iP_i, Q'] = Q$, while gauging $Q$ and $Q'$.

We formulate effective field theories associated with the spontaneous breaking of nonunifom 1-form symmetries. The resulting theories consist of multiple Abelian gauge fields [23, 24] combined in a specific manner dictated by the algebra. The configurations of the line operators in these theories are restricted (i. e. we have fractonic particles), and the mobility restriction is controlled by the commutation relations of charges with translations. For example, we can choose the algebras to realize theories whose low-energy limits are the scalar/vector charge gauge theories. The gapless excitations in these theories are understood as Nambu–Goldstone modes associated with the spontaneously broken 1-form symmetries. Owing to the nonuniform nature of the symmetry, some of the would-be Nambu–Goldstone modes acquire a gap, and the number of gapless modes is in general smaller than in the case where all 1-form charges commute with translations. This is the higher-form version of the inverse Higgs phenomenon [25–27] known in the case of spontaneously broken spacetime symmetries. Analogously to Maxwell theory, the resulting theories have electric and magnetic higher-form symmetries. These have 't Hooft anomalies, for which we identify the corresponding actions of Symmetry Protected Topological (SPT) phases in one higher dimension.

We can thus understand the appearance of gapless fractonic phases within the extension of Landau's symmetry-breaking paradigm to higher-form symmetries [28]. There have been other approaches to understanding fractonic phases using a symmetry principle: by considering certain exotic higher-form symmetries [29, 30], subsystem symmetries [31], or global symmetries which act on quasiparticles with position-dependent charge [32]. We emphasize that nonuniform higher-form symmetries in fact fall within the standard definition of higher-form symmetries, as the noncommutativity of the charge with translations does not mean that the symmetry generators are not topological. Rather, the topological property holds as it is a consequence of local conservation laws, as we discuss in Sec. 2.3.

The rest of this article is structured as follows. In Sec. 2, we introduce nonuniform symmetries and outline the strategy to realize fractonic phases based on them. In Sec. 3, we take a look at the theory with a $U(1)$ 1-form symmetry and a dipole 1-form symmetry, which re-

produces the scalar charge gauge theory at low energies. We discuss its various properties such as the types of line operators and their fractonic behavior, stability of the broken phase, and 't Hooft anomaly and SPT action. We also show that if we Higgs the theory to $\mathbb{Z}_n$ and restrict the possible configurations of dipole Wilson lines, we obtain the foliated field theory of the X-cube fracton order. In Sec. 4, we introduce a theory that at low energies reduces to the vector charge gauge theory, and again discuss its line operators and 't Hooft anomaly; it will be shown that this theory contains lineon defects. In Sec. 5, we discuss the generalization of the construction to the gauging of higher-pole symmetries. Section 6 is devoted to a summary and discussions. In Appendix A, we discuss the coupling of $U(1)$ and dipole gauge fields to a model of a complex scalar field. In Appendix B, we describe the evaluation of order parameters for higher-form symmetries in the theory described in Sec. 3. In Appendix C, we present a computation of the transformation properties of dipole Wilson line operators.

Before closing this section, let us summarize the notation used in this paper. In the following, $d$ denotes the spatial dimension, while $D = d + 1$ is the spacetime dimension. We work in flat Minkowski spacetime, and we use the mostly-plus convention for the Minkowski metric, $\eta_{\mu\nu} = (-1, +1, \ldots, +1)$. The Greek alphabet, $\mu, \nu \cdots$, will be used for spacetime indices, and the Latin alphabet, $i, j, k, \cdots$, will be used for spatial indices. The symmetrization and anti-symmetrization of indices are denoted by brackets, $(\cdots)$ and $[\cdots]$, respectively. For example, $A_{(ij)} := (A_{ij} + A_{ji})/2$ and $A_{[ij]} := (A_{ij} - A_{ji})/2$.

## 2 Fractons from nonuniform higher-form symmetries

In this section, we introduce the notion of nonuniform higher-form symmetries, and we outline the strategy to realize fractonic phases based on the spontaneous breaking of those symmetries.

### 2.1 Nonuniform higher-form symmetries

In quantum mechanics, a symmetry is a set of transformations of the rays of a Hilbert space that preserve the inner product and commute with the Hamiltonian. When we consider a quantum field theory, this definition is too general, since it allows for non-local operations. At the same time, the definition is also not sufficient because it does not accommodate spacetime symmetries; for example, the generator of a Lorentz boost does not commute with the Hamiltonian. In a quantum field theory, we usually employ a definition of symmetries compatible with the principle of locality. One such definition of symmetries that incorporates extended charged objects is higher-form symmetries [22].

Let us recall the definition of a higher-form symmetry [22]. We consider a quantum field theory (QFT) on a $D(= d + 1)$-dimensional spacetime manifold $X$. A QFT is said to have a $p$-form symmetry under a group $G$ when there exists an operator $U_g(M_{d-p})$, defined on a closed $(d-p)$-manifold $M_{d-p} \subset X$ and labeled by $g \in G$, such that:

- $U_g(M_{d-p})$ satisfies the group multiplication law,

$$U_{g_1}(M_{d-p}) U_{g_2}(M_{d-p}) = U_{g_1 g_2}(M_{d-p}), \quad \text{for } g_1, g_2 \in G, \tag{1}$$

- the dependence of $U_g(M_{d-p})$ on $M_{d-p}$ is *topological*, meaning that correlation functions that include this operator are unchanged under continuous deformations of $M_{d-p}$, unless the deformation crosses charged operators under the symmetry;

- there exists a set of charged operators $\{W_\alpha(C_p)\}$, defined on a closed $p$-manifold $C_p \subset X$, which are transformed by $U_g(M_{d-p})$ nontrivially. If $M_{d-p}$ is linked with $C_p$ once, the

transformation is given by

$$U_g(M_{d-p})W_\alpha(C_p) = R_g \cdot W_\alpha(C_p),\qquad(2)$$

where $R_g$ is a faithful representation of $G$ (meaning that for any $g \in G$ there is an operator $W_\alpha(C_p)$ such that $R_g \neq 1$).

The operator $U_g(M_{d-p})$ is called a symmetry generator. In the case of canonical quantization, $M_{d-p}$ and $C_p$ are chosen inside a spatial slice $V$. The symmetry action is expressed by the equal-time commutation relation

$$U_g(M_{d-p})W_\alpha(C_p)U_g^\dagger(M_{d-p}) = (R_g)^{I(M_{d-p},C_p)} \cdot W_\alpha(C_p),\qquad(3)$$

where $I(M,C)$ is the intersection number of $M$ and $C$.

In what follows we will mostly discuss continuous symmetries. For a continuous symmetry, there exists a conserved charge,

$$Q(M_{d-p}) := \int_{M_{d-p}} \star j,\qquad(4)$$

where $j$ is a $(p+1)$-form current density and $\star$ is the Hodge dual operation. The symmetry generator can be written as an exponential of a charge operator,

$$U(M_{d-p}) = \exp\left(i\alpha Q(M_{d-p})\right).\qquad(5)$$

We describe a continuous $p$-form symmetry as *uniform* if the action of a translation in any direction on the components of the $(p+1)$-form current density $j = \frac{1}{(p+1)!}j(x)_{\mu_1\cdots\mu_{p+1}}dx^{\mu_1}\wedge\cdots\wedge dx^{\mu_{p+1}}$ is given by a derivative,

$$[iP_\nu, j(x)_{\mu_1\cdots\mu_{p+1}}] = \partial_\nu j(x)_{\mu_1\cdots\mu_{p+1}}.\qquad(6)$$

Here we are interested in symmetries that do not satisfy this property. Specifically, we describe symmetries whose charges do no commute with translations, $[iP_\nu, Q] \neq 0$, as *nonuniform higher-form symmetries*.

Let us illustrate that, if a current satisfies Eq. (6), the corresponding charge operator $Q(M_{d-p}) = \int_{M_{d-p}} \star j$ (and hence the symmetry generator $U_g(M_{d-p})$) always commutes with $P_\mu$. This follows from the fact that for a conserved current $j$ we have

$$\int_{M_{d-p}} \partial_\mu \star j = 0,\qquad(7)$$

where $M_{d-p}$ is a $(d-p)$-cycle. To see this, note that the action of $\partial_\mu$ can be written as a Lie derivative, $\mathcal{L}_{e^\mu}$, along a constant vector field in the $\mu$-th direction, $e^\mu$. Thus,

$$\int_{M_{d-p}} \partial_\mu \star j = \int_{M_{d-p}} \mathcal{L}_{e^\mu} \star j = \int_{M_{d-p}} (i_{e^\mu}d \star j + di_{e^\mu} \star j),\qquad(8)$$

where $i_{e^\mu}$ is the interior product along $e^\mu$, and we have used Cartan's formula, $\mathcal{L}_\beta = i_\beta d + di_\beta$, for a given vector field $\beta$.[1] The first term vanishes due to the conservation law $d \star j = 0$, while

---

[1]Note that in this work we are considering only flat Minkowski spacetime, with spatial submanifolds aligned with the global Euclidean coordinates. For a curved manifold on a general background, the first equality of Eq. (8) would generalize to the covariant expression

$$\int_{M_{d-p}} e^\mu \nabla_\mu \star j = \int_{M_{d-p}} \mathcal{L}_e \star j,\qquad(9)$$

which holds for any vector field $e := e^\mu \partial_\mu$ that is covariantly constant, i.e. $\nabla_\rho e^\mu = 0$, such that $e^\mu$ defines an isometry.



Wilson line of charge $Q$

Gauging $Q$ and $Q'$
(SSB of $Q$ and $Q'$ 1-form symmetries)

$i$-th direction

$$[iP_i, Q'] = Q$$

Figure 1: Schematic illustration of the strategy to realize fractons. To render the worldline of charge $Q$ fractonic in the $i$-th direction, we gauge a charge $Q'$ which does not commute with $P_i$. In the resulting gauge theory, the Wilson line of charge $Q$ is confined to a constant-$x_i$ plane, implying that the particle cannot move in the $i$-th direction.

the second term also vanishes since it is a total derivative and $M_{d-p}$ is a $(d-p)$-cycle. The property (7) will be used repeatedly throughout this paper.

The consideration above implies that, if a conserved charge is to have a nontrivial commutation relation with translations, the corresponding current does not satisfy Eq. (6).

Immediate examples of nonuniform 0-form symmetries are spacetime symmetries such as rotations and boosts (Lorentzian/Carrollian/Galilean). Another example is multipole symmetries [14], which are also 0-form nonuniform symmetries. We will see that the spontaneous breaking of nonuniform higher-form symmetries can be used to construct (gapless) fractonic phases systematically.

## 2.2 Strategy to realize fractons

Let us outline the basic concept of how to realize fractonic phases via the spontaneous breaking of nonuniform higher-form symmetries (see Fig. 1 for a schematic illustration of the general idea).

As a first step, we interpret the existence of fractonic particles as a restriction on the possible configurations of worldlines of particles. Thus, it is natural to consider a theory with 1-form symmetries, whose charged objects can be regarded as worldlines. Suppose that, given a particle charged under $Q$, we would like to make it immobile in the $i$-th direction. This can be achieved by introducing another charge $Q'$ such that[2]

$$[iP_i, Q'] = Q, \quad \text{for a fixed direction } i. \tag{10}$$

In the terminology introduced earlier, the symmetry generated by $Q'$ is nonuniform. A theory with fractonic worldlines can be obtained by gauging $Q$ and $Q'$, by which we mean constructing the theory of dynamical gauge fields for $Q$ and $Q'$. The resulting theory exhibits the spontaneous breaking of 1-form symmetries corresponding to $Q$ and $Q'$.

Let us illustrate the procedure more concretely. Suppose the system has continuous Abelian 0-form symmetries generated by $Q$ and $Q'$ satisfying Eq. (10). We denote the corresponding 1-form currents as $j$ and $K$, in terms of which the charges are written as

$$Q(V) = \int_V \star j, \qquad Q'(V) = \int_V \star K, \tag{11}$$

---

[2]Here we took the translation to be in a spatial direction. If we introduce a charge that does not commute with $P_0$, we can realize a theory with "temporal fractons", meaning that their worldlines are confined in constant time slices.

where $V$ is a $d$-cycle. If we are to reproduce the algebra (10), the current for $Q'$ should be of the form

$$\star K = \star k - x_i \star j, \tag{12}$$

where $k$ is a uniform (but nonconserved) current. Explicitly, noting that the translation operator $P_j$ acts as a derivative on quantum fields and does not act on explicit coordinates (since commutation relations are among fields), we have

$$
\begin{aligned}
[iP_j, Q'(V)] &= \int_V \left[ \partial_j(\star k) - x_i(\partial_j \star j) \right] \\
&= \int_V \partial_j(\star k - x_i \star j) + \int_V (\partial_j x_i) \star j \\
&= \begin{cases} Q(V), & \text{for } j = i, \\ 0, & \text{otherwise,} \end{cases}
\end{aligned}
\tag{13}
$$

where we have used the property (7). Thus, the algebra (10) is reproduced. In this way, the consequence of the nonuniformity of the symmetry is that the corresponding currents should be related in a specific way with an explicit coordinate dependence (see Ref. [33]). The conservation laws of $Q$ and $Q'$ are

$$\mathrm{d} \star j = 0, \qquad \mathrm{d} \star K = \mathrm{d} \star k - \mathrm{d}x_i \wedge \star j = 0. \tag{14}$$

Denoting the gauge fields that couple to $Q$ and $Q'$ as $a = a_\mu \mathrm{d}x^\mu$ and $a' = a'_\mu \mathrm{d}x^\mu$, respectively, we introduce the coupling via a Lagrangian

$$\mathcal{L}_{\text{cpl}} = a \wedge \star j + a' \wedge \star k. \tag{15}$$

Note that we have coupled the gauge field $a'$ to the uniform and nonconserved part, $\star k$, rather than the conserved current, $\star K$. To implement the conservation laws (14), we postulate invariance under the gauge transformations

$$\delta a = \mathrm{d}\lambda + \sigma \mathrm{d}x_i, \qquad \delta a' = \mathrm{d}\sigma, \tag{16}$$

where $\lambda$ and $\sigma$ are 0-form gauge parameters for $Q$ and $Q'$, respectively.

Now consider the gauge theory where $a$ and $a'$ are dynamical (i.e. path-integrated) gauge fields. This gauge theory has 1-form symmetries: let us denote their generators by $Q(S)$ and $Q'(S)$,[3] where $S$ is a $(d-1)$-cycle. As we will see in examples in later sections, these symmetries are spontaneously broken. The generators inherit the nontrivial commutation relations with translations and satisfy[4]

$$[iP_i, Q'(S)] = Q(S). \tag{17}$$

The corresponding charged operators are the Wilson lines of $a$ and $a'$. In particular, the Wilson line operator of $a$, $W_q(C) = \mathrm{e}^{\mathrm{i}q \int_C a}$ where $C$ is a 1-cycle, is transformed under a gauge transformation as

$$W_q(C) \mapsto \mathrm{e}^{\mathrm{i}q \int_C \sigma \mathrm{d}x_i} W_q(C). \tag{18}$$

This is gauge-invariant only when the trajectory $C$ is confined to a plane with constant $x_i$,[5] which means that the particle cannot move in the $i$-th direction. Namely, the mobility of a

---

[3]We will use the same symbol $Q$ to denote the charge of a 0-form symmetry and the charge of the corresponding 1-form symmetry that appears as a result of the gauging of the former, to emphasize the connection between these two symmetries. When we wish to highlight the degree of the symmetry, we explicitly write the dependence on the underlying manifold over which the charge density is integrated, e.g. $Q(V)$ and $Q(S)$, where $V$ and $S$ are a $d$-cycle and a $(d-1)$-cycle, respectively.

[4]Since $p$-form symmetries are Abelian if $p > 0$, $Q$ and $Q'$ should be Abelian charges in this construction.

[5]The time direction is also allowed.

particle charged under $Q$ is restricted. If we also wish to make the particle immobile in the $j$-th direction, we can add another charge $Q''$ whose commutator with $P_j$ becomes $Q$. In this way, we can control the mobility of a particle by choosing an algebra with spatial translational generators.

For example, if we consider 1-form symmetries of charges $Q$ and $\{Q_i\}_{i=1\ldots d}$ whose commutation relations with $P_i$ are given by

$$[iP_i, Q_j] = \delta_{ij} Q, \quad \text{for any } i, j \in \{1, \ldots, d\}, \tag{19}$$

a particle charged under $Q$ becomes immobile in every spatial direction, meaning that it is a fracton. The gauge theory of charges $Q$ and $Q_i$ is the coupled vector gauge theory discussed in Refs. [23, 24, 34]. In the low-energy limit, this theory reduces to the scalar charge gauge theory [6, 7]. If we further wish to make the charges $Q_i$ immobile, we can introduce an additional set of charges, $\{Q_{ij}\}_{i,j=1\ldots d}$, where $Q_{ij} = Q_{ji}$, satisfying the algebra

$$[iP_k, Q_{ij}] = \delta_{ki} Q_j + \delta_{kj} Q_i, \tag{20}$$

and also gauge $Q_{ij}$. Then, the worldlines of particles charged under $Q_i$ can be made completely fractonic.

As another example, we can take a set of charges, $\{q_i, Q_i\}_{i=1\ldots d}$ with $d = 3$, whose commutation relations with $P_i$ are given by

$$[iP_i, Q_j] = \epsilon_{ijk} q_k, \qquad [iP_i, q_j] = 0. \tag{21}$$

Then a specific charge, for example $q_z$, can be written as a commutator of a translation and another charge,

$$q_z = [iP_x, Q_y] = -[iP_y, Q_x]. \tag{22}$$

This implies that a particle with charge $q_z$ cannot move in the $x$ and $y$ directions. Thus, a particle with a charge vector $\boldsymbol{q}$ can only move in the direction of $\boldsymbol{q}$, and is a lineon. The gauge theory is described by the corresponding gauge fields for $Q_i$ and $q_i$. This theory reduces to the vector charge gauge theory [7] at low energies.

A gauge theory constructed this way has magnetic symmetries that are also nonuniform. In (3+1) dimensions, the magnetic symmetries are also 1-form symmetries, and the corresponding charged objects are the worldlines of magnetic monopoles. As a result of the nonuniformity of the magnetic symmetries, some of the magnetic monopoles also become fractonic. More concretely, in the gauge theory based on $[iP_i, Q'] = Q$, there are magnetic monopoles for $Q$ and $Q'$ if their symmetry group is $U(1)$ rather than $\mathbb{R}$, and their worldlines are the charged objects of magnetic 1-form symmetries. The gauge-invariant field strengths in this example are

$$f = da - a' \wedge dx^i, \qquad f' = da', \tag{23}$$

and their Bianchi identities are

$$df = -f' \wedge dx_i, \qquad df' = 0. \tag{24}$$

The Bianchi identity for $f$ can be written in the form of a conservation law as

$$d(f + x_i f') = 0. \tag{25}$$

The conserved charges of magnetic symmetries are given by

$$Q^{\mathrm{m}}(S) = \frac{1}{2\pi} \int_S (f + x_i f'), \qquad Q'^{\mathrm{m}}(S) = \frac{1}{2\pi} \int_S f', \tag{26}$$

where $S$ is a 2-cycle. One can see that the magnetic charges satisfy

$$[iP_i, -Q^{\mathrm{m}}] = Q'^{\mathrm{m}}. \tag{27}$$

Note that the positions of $Q$ and $Q'$ are swapped relative to the original algebra. The magnetic algebra (27) implies that the magnetic monopole of $Q'$ cannot move in the $i$-th direction. In subsequent sections we will demonstrate this more explicitly through examples.

The existence of magnetic nonuniform symmetries explains the presence of fractons in the theory of elasticity in $(2+1)$ dimensions [3, 4, 6, 7]. The relevant symmetries are spatial translations and rotations. The generators of translations $P_i$ and the rotation $L$ satisfy $[iP_i, L] = -\epsilon_{ij} P_j$, which can be written explicitly as

$$[iP_x, L] = -P_y, \qquad [iP_y, L] = P_x. \tag{28}$$

Upon the formation of a solid, the translational and rotational symmetries, which are 0-form symmetries, are both spontaneously broken. As a result, there are magnetic 1-form symmetries, and the corresponding charged objects are the worldlines of dislocations for $P_i$ and disclinations for $L$, respectively. The charges of magnetic symmetries, $P_i^{\mathrm{m}}$[6] and $L^{\mathrm{m}}$, satisfy the following algebra,

$$[iP_x, P_y^{\mathrm{m}}] = L^{\mathrm{m}}, \qquad [iP_y, -P_x^{\mathrm{m}}] = L^{\mathrm{m}}. \tag{29}$$

These relations imply that disclinations, which are charged under $L^{\mathrm{m}}$, are immobile in both the $x$ and $y$ directions and are therefore fractons. Thus, the fractonic behavior of disclinations is a consequence of the nonuniform magnetic symmetries, which arise from the spontaneous breaking of nonuniform 0-form symmetries.

In this way, the mobility restrictions of particles/monopoles are fully controlled in the present construction by the commutation relations with translations, and we can implement the restrictions as we wish by modifying the underlying algebra. This method reproduces various gapless fractonic models, and helps us to engineer systems with desired fractonic properties. Some general features of the gauge theories constructed in this manner are summarized as follows.

- Due to the nonuniformity of the symmetry, the corresponding current has an explicit coordinate dependence whose particular structure is dictated by the algebra.

- The 1-form symmetries are spontaneously broken [35]. The gapless modes in such theories are the Nambu–Goldstone modes associated with 1-form symmetry breaking.

- Some of the Nambu–Goldstone modes acquire a gap, because of the nonuniform nature of the broken symmetries. This is the higher-form version of the so-called inverse Higgs mechanism [25–27] for spacetime symmetries.

- If the spontaneously broken 1-form symmetries are compact (i.e. $U(1)$, not $\mathbb{R}$), there are emergent magnetic symmetries that are also nonuniform. In $(3+1)$ dimensions, the magnetic symmetries are 1-form symmetries, and the worldlines of magnetic monopoles are the corresponding charged objects. Due to the nonuniform nature of the symmetry, certain magnetic monopoles become fractonic.

- Similarly to the case of Maxwell theory, there is an 't Hooft anomaly between electric and magnetic symmetries. We can identify the action of the bulk Symmetry-Protected Topological (SPT) phase to match the anomaly.

---

[6]The magnetic symmetry for a translation $P_i$ appears because the order parameter space for one direction of translations is $\mathbb{R}/\mathbb{Z} \simeq U(1)$.

## 2.3 Deformation of symmetry generators

As we mentioned earlier, the fact that the conserved charges of nonuniform symmetries do not commute with spacetime translations $P_\mu$ does not mean that the symmetry generators are not topological. Even if a charge does not commute with $P_\mu$, the symmetry generator remains topological as long as the local conservation law is satisfied. Thus, a nonuniform higher-form symmetry indeed falls under the umbrella of higher-form symmetries as per the definition in Sec. 2.1 and does not require anything further. In this subsection, we demonstrate this fact.

Suppose a QFT has a $p$-form continuous symmetry whose $(p+1)$-form current density is $j$. The charge operator is given by the integral of $\star j$ over a closed submanifold. For a given smooth embedding $\phi : S \to X$, where $S$ is a $(d-p)$-dimensional closed manifold, the conserved charge can be written as the integral of the pullback of $\star j$ by $\phi$,

$$Q(S,\phi) = \int_S \phi^*(\star j) = \int_{\phi(S)} \star j. \tag{30}$$

The map $\phi$ can be denoted by $x^\mu(s^a) \in X$, where $s^a$ is a coordinate of $S$.

Let us consider local deformations of the symmetry generators. Writing the conserved charge as

$$Q(S,\phi) = \int_S \phi^*(\star j) = \int_S \phi^*\left[\frac{1}{(d-p)!}(\star j)_{\mu_1\mu_2\cdots\mu_{d-p}}(x)\mathrm{d}x^{\mu_1}\wedge\cdots\wedge\mathrm{d}x^{\mu_{d-p}}\right], \tag{31}$$

we may perform a local deformation of the manifold along the vector field $\beta^\mu(x)$,

$$x^\mu(s^a) \mapsto x^\mu(s^a) + \epsilon\beta^\mu(x(s)), \tag{32}$$

where $\epsilon$ is an infinitesimal parameter. Under an infinitesimal deformation along $\beta^\mu$, the change of the charge operator is given by the Lie derivative of $\star j$,

$$\begin{aligned}
\delta_{\epsilon\beta}Q(S,\phi) &= \frac{1}{(d-p)!}\int_S \phi^*\Big[(\star j)_{\mu_1\cdots\mu_{d-p}}(x+\epsilon\beta)\mathrm{d}(x+\epsilon\beta)^{\mu_1}\wedge\cdots\wedge\mathrm{d}(x+\epsilon\beta)^{\mu_{d-p}} \\
&\qquad\qquad -(\star j)_{\mu_1\cdots\mu_{d-p}}(x)\mathrm{d}x^{\mu_1}\wedge\cdots\wedge\mathrm{d}x^{\mu_{d-p}}\Big] \\
&= \epsilon\int_S \phi^*\mathcal{L}_\beta\star j + O(\epsilon^2) \\
&= \epsilon\int_S \phi^*\big(i_\beta\mathrm{d}\star j + \mathrm{d}\,i_\beta\star j\big) + O(\epsilon^2).
\end{aligned} \tag{33}$$

If the current satisfies a local conservation law, $\mathrm{d}\star j = 0$, the first term vanishes. Moreover, since $S$ is closed, the second term also vanishes. By this argument, even if the charge does not commute with $P_\mu$, the symmetry generator is topological as long as the current is locally conserved.

In fact, even if the symmetry is uniform, we cannot perform a local deformation of a generator using the energy-momentum tensor. To see this, consider a transformation by an operator $e^{i\int_V \epsilon\beta^\mu(x)p_\mu(x)}$, where $p_\mu(x)$ is the energy-momentum density at $x$. For a uniform symmetry, the action of this operator on the current density is

$$\begin{aligned}
&e^{i\int_V \epsilon\beta^\mu(y)p_\mu(y)}\left[\frac{1}{(d-p)!}(\star j)_{\mu_1\cdots\mu_{d-p}}(x)\mathrm{d}x^{\mu_1}\wedge\cdots\wedge\mathrm{d}x^{\mu_{d-p}}\right]e^{-i\int_V \epsilon\beta^\mu(y)p_\mu(y)} \\
&= \frac{1}{(d-p)!}(\star j)_{\mu_1\cdots\mu_{d-p}}(x+\epsilon\beta(x))\mathrm{d}x^{\mu_1}\wedge\cdots\wedge\mathrm{d}x^{\mu_{d-p}}.
\end{aligned} \tag{34}$$

Since the commutator can act only on quantum fields, $p_\mu(x)$ cannot translate the measure of the spacetime integral. For a nonuniform symmetry, the current depends explicitly on the coordinates, and $p_\mu(x)$ cannot generate translations of the current density. In this way, the commutativity of a symmetry generator with $P_\mu$ is unrelated to the topological nature of the generator of continuous symmetries, since this property is a consequence of the local conservation law.

# 3  Gauge theory with $U(1)$ and dipole symmetries

Here we discuss a gauge theory with a $U(1)$ symmetry and a dipole symmetry. Starting from the symmetry algebra, we construct a theory containing dynamical $U(1)$ and dipole gauge fields, in which the Wilson line of a $U(1)$ charge is fractonic. This theory reduces to the scalar charge gauge theory at low energies.

## 3.1  $U(1)$ and dipole symmetries and their gauging

Suppose the action is invariant under two kinds of 0-form symmetries, generated by $Q$ and $\{Q_i\}_{i=1\dots d}$, respectively. We will refer to the former as the $U(1)$ charge symmetry, and the latter as the dipole symmetry.[7] These charges are Abelian and commute with each other. The $U(1)$ charge symmetry commutes with all other generators, while the dipole symmetry is characterized by its commutation relation with spatial translations $P_i$:[8]

$$[iP_i, Q_j] = -\delta_{ij}Q, \qquad [iP_i, Q] = 0. \tag{36}$$

Noether's theorem implies the existence of conserved 1-form currents, $j$ and $K_i$, for the $U(1)$ charge symmetry and the dipole symmetry, respectively, whose conservation laws read

$$d \star j = 0, \qquad d \star K_i = 0. \tag{37}$$

The conserved charges can be written as

$$Q(V) := \int_V \star j, \qquad Q_i(V) := \int_V \star K_i, \tag{38}$$

where $V$ is a $d$-cycle.

As a consequence of the algebra (36), the dipole current $K_i$ is nonuniform, while the $U(1)$ charge current $j$ is uniform. The nonuniformity of the dipole current results in certain relations between the currents, as discussed in Ref. [33]. To reproduce this algebra, the dipole current should take the form

$$\star K_i = \star k_i + x_i \star j, \tag{39}$$

where the current $k_i$ is uniform but not conserved. The conservation law of the dipole current reads

$$d \star K_i = d \star k_i + dx_i \wedge \star j = 0. \tag{40}$$

---

[7]We here consider a generic theory with there symmetries and discuss its coupling to gauge fields. As a concrete model, we can consider a theory of a complex scalar field that is transformed under $U(1)$ charge- and dipole-transformations as

$$\Phi \mapsto e^{i(\alpha + \beta^i x_i)}\Phi, \tag{35}$$

where $\alpha$ and $\beta_i$ are constant parameters. We detail on the coupling of gauge fields this model in Appendix A.

[8]While it may be natural to have a plus sign on the right-hand side of Eq. (36), here we chose a minus sign to allow for easier comparison with prior works, for example, Ref. [16]. The properties of the theory are unchanged by this convention.

We would now like to gauge these symmetries.[9] In doing so, we would like to respect the relation (39), which comes from the underlying algebra. For this purpose, rather than introducing background gauge fields for conserved currents $(j, K_i)$, we introduce gauge fields for the currents $(j, k_i)$ as

$$S_{\mathrm{cpl}} = \int_X \left( a \wedge \star j + \mathcal{A}_i \wedge \star k^i \right), \tag{41}$$

where $a = a_\mu \mathrm{d}x^\mu$ and $\mathcal{A}_i = (\mathcal{A}_i)_\mu \mathrm{d}x^\mu$ are 1-form gauge fields. To implement the conservation laws (37), we require invariance under the following gauge transformations,

$$\delta a = \mathrm{d}\Lambda - \Sigma_i \mathrm{d}x^i, \qquad \delta \mathcal{A}_i = \mathrm{d}\Sigma_i, \tag{42}$$

where $\Lambda$ and $\Sigma_i$ are the gauge parameters for the $U(1)$ and dipole gauge transformations, respectively. Indeed, requiring that the action is invariant under the dipole gauge transformation,

$$0 = \delta S_{\mathrm{cpl}} = \int_X \left( -\Sigma_i \mathrm{d}x^i \wedge \star j + \mathrm{d}\Sigma_i \wedge \star k^i \right) = -\int_X \Sigma_i \left( \mathrm{d}x^i \wedge \star j + \mathrm{d} \star k^i \right), \tag{43}$$

we find that dipole gauge invariance implies Eq. (40). Importantly, the spatial part of the $U(1)$ gauge field is transformed by a dipole gauge transformation (42), as a consequence of the nonuniform nature of the dipole symmetry. Such a shift of gauge fields is known to occur in the presence of higher group symmetries [38].

## 3.2 Gauge theory of $U(1)$ charge- and dipole-gauge fields

Now we promote the gauge fields $(a, \mathcal{A}_i)$ to dynamical fields and consider their gauge theory.[10] This gauge theory can be regarded as the theory of Nambu–Goldstone bosons resulting from spontaneously broken 1-form symmetries.[11] The building blocks of the theory are the gauge-invariant field strengths

$$f := \mathrm{d}a + \mathcal{A}_i \wedge \mathrm{d}x^i, \qquad \mathcal{F}_i := \mathrm{d}\mathcal{A}_i. \tag{44}$$

Since the charges $Q$ and $Q_i$ are both Abelian, we can take the corresponding symmetry group to be either $\mathbb{R}$ or $U(1)$. While we take the symmetry of $Q$ to be $U(1)$, we consider both $\mathbb{R}$ and $U(1)$ for the dipole symmetry group. Although this choice does not affect the local properties of the theory, such as the number of gapless modes, it leads to a difference in the identification of extended operators.

The choice of $U(1)$ or $\mathbb{R}$ affects the normalization of the gauge fields. We normalize the $U(1)$ gauge field $a$ in order to satisfy the Dirac quantization condition

$$\int_S \mathrm{d}a \in 2\pi\mathbb{Z}, \tag{45}$$

where $S$ is a 2-cycle. In the normalization condition of $a$, "$\mathrm{d}a$" is a local expression, so it is implicit in the expression above that we should combine the coordinate patches for noncontractible manifolds. Similarly, the gauge parameter $\Lambda$ satisfies

$$\int_C \mathrm{d}\Lambda \in 2\pi\mathbb{Z}, \tag{46}$$

where $C$ is a 1-cycle.

---

[9]See also Refs. [16, 36, 37].

[10]See Ref. [39] for a related construction.

[11]For uniform higher-form symmetries, the field strengths appear as Maurer–Cartan forms, with which invariant effective Lagrangians can be constructed [40].

We should also specify normalization conditions for the dipole gauge fields and gauge parameters. In the case where we take the dipole symmetry group to be $U(1)$, we employ the normalization

$$\ell \int_S \mathcal{F}_i \in 2\pi\mathbb{Z}, \qquad \ell \int_C d\Sigma_i \in 2\pi\mathbb{Z}, \tag{47}$$

where $\ell$ is a parameter with dimensions of length. On the other hand, when we consider $\mathbb{R}$, these integrals are taken to be zero. Note that the normalization condition of $a$ is not affected by dipole gauge transformations. Under such a transformation,

$$\delta \int_S da = - \int_S d\Sigma_i \wedge dx^i. \tag{48}$$

For example, let us evaluate this on a 2-sphere. By expressing a 2-sphere as a union of northern and southern hemispheres, $S^2 = S^+ \cup S^-$, the integral on the right-hand side of (48) can be written as

$$\int_{S^+} d\Sigma_i^+ \wedge dx^i + \int_{S^-} d\Sigma_i^- \wedge dx^i = \int_{S^1} \left( \Sigma_i^+ - \Sigma_i^- \right) dx^i = \frac{1}{\ell} \int_{S^1} 2\pi n_i dx^i = 0, \tag{49}$$

where $S^1$ is the equator, and we used the fact that $\Sigma_i^+(x) = \Sigma^-(x) + 2\pi n_i/\ell$, where $n_i$ are integers.

We can construct effective theories using the gauge-invariant field strengths (44). To this end, we shall employ the effective action[12]

$$S[a, \mathcal{A}_i] = \int_X \left( -\frac{1}{2e^2} f \wedge \star f - \frac{1}{2(e_1)^2} \mathcal{F}_i \wedge \star \mathcal{F}_i \right). \tag{51}$$

Since we do not enforce Lorentz symmetry, the coefficients of the time and spatial derivative terms may be different in general. However, in the following we take them to be the same for notational simplicity — a more general choice of coefficients does not affect our conclusions.[13] The equations of motion resulting from the action (51) as well as the Bianchi identities for $f$ and $\mathcal{F}_i$ are summarized as

$$\frac{1}{e^2} d \star f = 0, \tag{53}$$

$$\frac{1}{(e_1)^2} d \star \mathcal{F}_i + \frac{1}{e^2} dx^i \wedge \star f = 0, \tag{54}$$

$$df - \mathcal{F}_i \wedge dx^i = 0, \tag{55}$$

$$d\mathcal{F}_i = 0. \tag{56}$$

---

[12]We can also add theta terms for the $U(1)$ gauge fields and dipole gauge fields (if the dipole symmetry group is $U(1)$),

$$S_\theta = \theta \int_X \frac{1}{2} \frac{da}{2\pi} \wedge \frac{da}{2\pi} + \sum_i \theta_i \ell^2 \int_X \frac{1}{2} \frac{\mathcal{F}_i}{2\pi} \wedge \frac{\mathcal{F}_i}{2\pi}, \tag{50}$$

where the summation over $i$ is denoted explicitly. The theta term for $a$ is gauge invariant up to a total derivative. The theta angles are $2\pi$ periodic on spin manifolds.

[13]One can further add terms with one derivative which are of the form $C_{ijk}\mathcal{A}_i \wedge \mathcal{F}_j \wedge dx_k$, where $C_{ijk}$ are coefficients. These are also gauge invariant up to a total derivative. The existence of such terms corresponds to the situation where the expectation values of the dipole 1-form charges become nonzero [40],

$$\langle [Q_i(S_1), Q_j(S_2)] \rangle \propto C_{ijk} \int_{S_1 \cap S_2} dx^k, \tag{52}$$

where $Q_i(S_1)$ is the generator of the dipole 1-form symmetry (see Sec. 3.5) and $S_1$ and $S_2$ are 2-cycles.

We may identify the canonical momenta for $a$ and $\mathcal{A}_i$ as

$$\pi := \frac{\mathrm{d}}{\mathrm{d}\mathrm{d}a}\mathcal{L} = -\frac{1}{e^2} \star f\,, \qquad \Pi_i := \frac{\mathrm{d}}{\mathrm{d}\mathrm{d}\mathcal{A}_i}\mathcal{L} = -\frac{1}{(e_1)^2} \star \mathcal{F}_i\,. \tag{57}$$

The canonical commutation relations can be written as

$$\left[\iint_S \pi, a\right] = \mathrm{i}\delta_V(S)\,, \qquad \left[\iint_S \Pi_i, \mathcal{A}_j\right] = \mathrm{i}\delta_{ij}\delta_V(S)\,, \tag{58}$$

where $S$ is a 2-cycle inside a spatial slice $V$ and $\delta_V(S)$ is the Poincaré dual of $S$ with respect to $V$.

### 3.3 Low energy limit and the relation to the scalar charge gauge theory

Let us consider the low-energy limit of the action (51). Since the gauge-invariant field strength $f$ involves $\mathcal{A}_i$, some components of $\mathcal{A}_i$ acquire a mass given by $m^2 := (e_1)^2/e^2$ and drop out at low energies. As a result, the number of gapless Nambu–Goldstone modes is reduced compared to the case of uniform symmetries [40]. This is the higher-form version of the so-called inverse Higgs phenomenon [25–27].

We introduce electric and magnetic fields for $\mathcal{F}_i$ and $\mathrm{d}a$ as

$$\mathcal{F}_i = (\mathcal{E}_i)_j \mathrm{d}x^j \mathrm{d}x^0 + \frac{1}{2}\epsilon_{jkl}(\mathcal{B}_i)^l \mathrm{d}x^j \mathrm{d}x^k\,, \tag{59}$$

$$\mathrm{d}a = e_i\,\mathrm{d}x^i \mathrm{d}x^0 + \frac{1}{2}\epsilon_{ijk}b^k \mathrm{d}x^i \mathrm{d}x^j\,. \tag{60}$$

Note that $e_i$ and $b_i$ are not invariant under dipole gauge transformations. Let us consider physics at energies well below the mass gap. The equation of motion for the gapped part of $\mathcal{A}_i$ reads

$$\mathrm{d}a + \mathcal{A}_j \wedge \mathrm{d}x^j = 0\,. \tag{61}$$

Thus, some components of the dipole gauge fields are expressed in terms of $a$ as

$$(\mathcal{A}_i)_0 = \partial_i a_0 - \partial_0 a_i = e_i\,, \tag{62}$$

$$(\mathcal{A}_{[i})_{j]} = \partial_{[i}a_{j]} = \frac{1}{2}\epsilon_{ijk}b_k\,. \tag{63}$$

Note that the quantities on both sides have the same transformation properties: they are $U(1)$ gauge invariant (the expressions on the right-hand side are $U(1)$ electric/magnetic fields) but not dipole gauge invariant. At low energies, the dynamical variables are reduced to

$$\{a_0, a_i, (\mathcal{A}_i)_0, (\mathcal{A}_i)_j\} \to \{a_0, a_i, (\mathcal{A}_{(i)j)}\}\,. \tag{64}$$

The low-energy limit of this theory can be connected to the scalar charge gauge theory as follows. We introduce a rank-2 symmetric tensor gauge field by

$$A_{ij} := (\mathcal{A}_{(i)j)} + \partial_{(i}a_{j)} = (\mathcal{A}_i)_j + \partial_j a_i\,, \tag{65}$$

where the equality on the right holds only at low energies where Eqs. (62) and (63) apply. The field $A_{ij}$ is dipole gauge invariant and is transformed by a $U(1)$ gauge transformation as

$$\delta A_{ij} = \partial_i \partial_j \Lambda\,. \tag{66}$$

The gauge fields, $\{a_0, A_{ij}\}$, constitute the fields that appear in the scalar charge gauge theory. Using Eqs. (62) and (63), the dipole electric and magnetic fields are written at low energies as

$$(\mathcal{E}_i)_j = \partial_j (\mathcal{A}_i)_0 - \partial_0 (\mathcal{A}_i)_j = \partial_j \partial_i a_0 - \partial_0 A_{ij}\,, \qquad \partial_{[j}(\mathcal{A}_i)_{k]} = \partial_{[j}A_{k]i}\,. \tag{67}$$

Thus, the Lagrangian density in the IR reads

$$\mathcal{L} = \frac{1}{2(e_1)^2}(\mathcal{E}_i)_j(\mathcal{E}_i)_j - \frac{1}{(e_1)^2}\partial_{[j}(\mathcal{A}_i)_{k]}\partial_{[j}(\mathcal{A}_i)_{k]}$$
$$= \frac{1}{2(e_1)^2}(\partial_0 A_{ij} - \partial_i\partial_j a_0)^2 - \frac{1}{(e_1)^2}\partial_{[j}A_{k]i}\partial_{[j}A_{k]i}. \tag{68}$$

Equation (68) is nothing but the Lagrangian of the scalar charge gauge theory.

The equations of motion for the scalar charge gauge theory can also be derived from Eqs. (53)–(56). Writing $f_{i0} = \widetilde{e}_i$ and $f_{ij} = \epsilon_{ijk}\widetilde{b}^k$, the independent components of Eq. (53) are

$$\partial_i\widetilde{e}_i = 0, \tag{69}$$

$$\partial_0\widetilde{e}_i - \epsilon_{ijk}\partial^j\widetilde{b}^k = 0. \tag{70}$$

Evaluating Eq. (54) explicitly gives

$$\widetilde{e}_i = -\frac{e^2}{(e_1)^2}\partial^j(\mathcal{E}_i)_j, \tag{71}$$

$$\epsilon_{ijk}\widetilde{b}^k = \frac{e^2}{(e_1)^2}\left[\partial_0(\mathcal{E}_j)_i - \epsilon_{ikl}\partial^k(\mathcal{B}_j)^l\right]. \tag{72}$$

Inserting Eq. (71) into Eq. (69) reveals the equation of motion

$$\partial^i\partial^j(\mathcal{E}_i)_j = 0. \tag{73}$$

This is valid at all energies, and in particular at low energies it matches the first equation of motion for the electric field in the scalar charge gauge theory. Furthermore, the symmetric part of Eq. (72) yields the second equation of motion,

$$\partial_0(\mathcal{E}_{(i})_{j)} - \epsilon_{kl(i}\partial^k(\mathcal{B}_{j)})^l = 0, \tag{74}$$

which similarly matches the low-energy dynamics given by the scalar charge gauge theory.[14] The third and fourth equations of motion also emerge from the explicit components of the Bianchi identity Eq. (56) as

$$\epsilon^{jkl}\partial_j(\mathcal{E}_i)_k + \partial_0(\mathcal{B}_i)^l = 0, \qquad \partial_l(\mathcal{B}_i)^l = 0, \tag{75}$$

again agreeing with the low-energy description. Finally, expanding to leading order in $e^2/(e1)^2$, the leading contribution to Eq. (55) is

$$\mathcal{F}_i \wedge dx^i = 0. \tag{76}$$

This implies that at low energies, the electric field is symmetric, $(\mathcal{E}_{[i})_{j]} = 0$, and the magnetic field is traceless, $(\mathcal{B}_i)^i = 0$. Thus, the full structure and dynamics of the scalar charge gauge theory are recovered in the low-energy limit.

## 3.4 Number of gapless modes

Let us discuss the number of gapless modes in this theory. The SSB of a 1-form symmetry gives $d-1$ physical gapless Nambu–Goldstone modes. There is one $U(1)$ 1-form symmetry corresponding to the charge $Q$, and $d$ $U(1)$ (or $\mathbb{R}$) 1-form symmetries for the dipole charges $Q_i$. Due to the inverse Higgsing, the antisymmetric components, $(\mathcal{A}_{[i})_{j]}$, are expressed in terms of

---

[14]Eq. (70) and the skew-symmetric part of Eq. (72) do not yield any new information: combining them using Eq. (71) simply produces the divergence of Eq. (74).

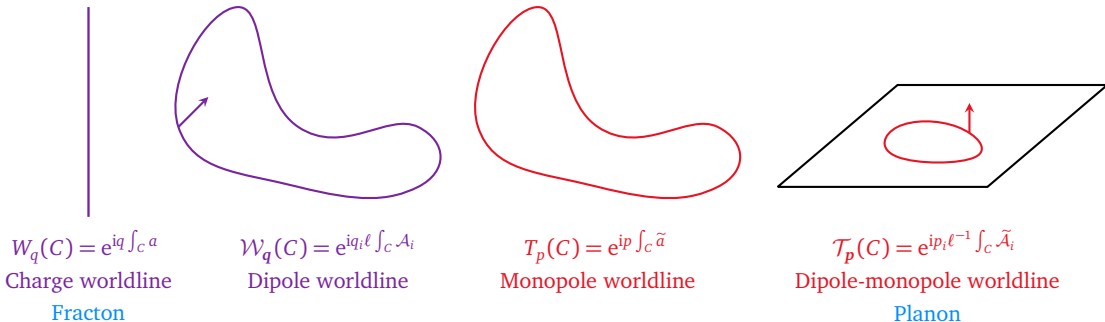

$$W_q(C) = e^{iq\int_C a} \qquad \mathcal{W}_q(C) = e^{iq_i\ell\int_C \mathcal{A}_i} \qquad T_p(C) = e^{ip\int_C \tilde{a}} \qquad \mathcal{T}_p(C) = e^{ip_i\ell^{-1}\int_C \tilde{\mathcal{A}}_i}$$

Charge worldline     Dipole worldline     Monopole worldline     Dipole-monopole worldline

Fracton                                             Planon

Figure 2: Summary of charged objects in this theory. Dipole operators are characterized by charge vectors, which are indicated by arrows. A $U(1)$ charge is a fracton. A dipole-monopole is a planon and it can only move perpendicularly to its charge vector. There is no restriction for the other two. Dipole monopoles are absent when the dipole symmetry group is chosen to be $\mathbb{R}$.

other degrees of freedom (note that the scalar potential part, $(\mathcal{A}_i)_0$, does not have a conjugate momentum and does not produce physical degrees of freedom). Thus, the total number of physical gapless modes is counted as

$$\overbrace{(d-1)}^{U(1)\text{ 1-form SSB}} + \overbrace{(d-1)\times d}^{\text{Dipole 1-form SSB}} \underbrace{-\frac{d(d-1)}{2}}_{\text{Inverse Higgsing}} = \overbrace{\frac{d(d+1)}{2}}^{\text{Sym. tensor of rank 2}} \underbrace{-1}_{\text{Gauss law}} . \tag{77}$$

This is consistent with the result for the scalar charge gauge theory in $d$-spatial dimensions [16], the counting from which is written on the right-hand side of Eq. (77). We can count the physical gapless modes in the scalar charge gauge theory as follows. The symmetric electric field, $E_{ij}$, has $d(d+1)/2$ components, and each component is gauge-invariant and produces one physical mode, except for the Gauss law constraint, $\partial_i\partial_j E_{ij} = 0$, which eliminates one mode and corresponds to $-1$.

This relation generalizes to other theories with spontaneously-broken nonuniform higher-form symmetries. In Sec. 4.6 this counting argument will be extended to theories with a vector of $U(1)$ charges, and subsequently generalized to theories obtained by gauging higher multipole moments in Sec. 5.3.

## 3.5 Higher-form symmetries and fractons

Let us identify the higher-form symmetries of this theory. See Fig. 2 for a summary of objects charged under the 1-form symmetries.

The charge operators of the $U(1)$ charge 1-form symmetry and the dipole 1-form symmetry are

$$Q(S) := \frac{1}{e^2}\int_S \star f = -\int_S \pi , \tag{78}$$

$$Q_i(S) := \int_S \left(\frac{1}{(e_1)^2} \star \mathcal{F}_i + \frac{1}{e^2} x_i \star f\right) = -\int_S \Pi_i - \int_S x_i \pi , \tag{79}$$

respectively, where $S$ is a $(d-1)$-cycle. If we place $S$ inside a spatial slice, $Q_i(S)$ can be regarded

as an operator that acts on the Hilbert space. It generates a nonuniform symmetry and satisfies

$$
\begin{aligned}
[\mathrm{i}P_i, Q_j(S)] &= \int_S \left( \frac{1}{(e_1)^2} \partial_i \star \mathcal{F}_j + \frac{1}{e^2} x_j \partial_i \star f \right) \\
&= \int_S \partial_i \left( \frac{1}{(e_1)^2} \star \mathcal{F}_j + \frac{1}{e^2} x_j \star f \right) - \delta_{ij} \frac{1}{e^2} \int_S \star f \\
&= -\delta_{ij} Q(S),
\end{aligned}
\tag{80}
$$

where we have used Eq. (7). Thus, the charges of 1-form symmetries reproduce the algebra (36).

The charged object for $Q(S)$ is the Wilson line of a $U(1)$ charge and it is fractonic, i.e. its worldline should be aligned straight along the time direction and cannot move in space. Let us illustrate this explicitly. The Wilson line is written as

$$
W_q(C) := \mathrm{e}^{\mathrm{i}q \int_C a}, \tag{81}
$$

where $C$ is a line in spacetime and $q \in \mathbb{Z}$. This is invariant under a $U(1)$ gauge transformation, but for an arbitrary choice of $C$ it is modified by a dipole gauge transformation,

$$
W_q(C) \mapsto \mathrm{e}^{-\mathrm{i}q \int_C \Sigma_i \mathrm{d}x_i} W_q(C). \tag{82}
$$

For this to be dipole-gauge-invariant, $C$ should be a straight line parallel to the time axis. Thus, a $U(1)$ charge, whose worldline is given by $C$, is fractonic. One can check that $Q(S)$ indeed generates a $U(1)$ phase rotation of $W_q(C)$,[15]

$$
\mathrm{e}^{\mathrm{i}\alpha Q(S)} W_q(C) = \mathrm{e}^{\mathrm{i}\alpha q L(S,C)} W_q(C), \tag{84}
$$

where $C$ is placed along the time direction and $L(S,C)$ is the linking number of $S$ and $C$ in 4-dimensional spacetime. We can also define another gauge-invariant operator with a trajectory of a $U(1)$ charge,

$$
W_q'(S_C) = \mathrm{e}^{\mathrm{i}q \int_C a + \mathrm{i}q \int_{S_C} \mathcal{A}_i \wedge \mathrm{d}x^i} = \mathrm{e}^{\mathrm{i}q \int_{S_C} (\mathrm{d}a + \mathcal{A}_i \wedge \mathrm{d}x^i)}, \tag{85}
$$

where $S_C$ is a surface whose boundary is $C$. Although $C$ can be of an arbitrary shape, this is not a line operator since a surface is attached. As the integrand is simply given by the field strength $f$, at low energies this operator becomes trivial.

We also have dipole Wilson lines,

$$
\mathcal{W}_{\boldsymbol{q}}(C) := \mathrm{e}^{\mathrm{i}q_i \ell \int_C \mathcal{A}_i}, \tag{86}
$$

where $q_i$ is a charge vector. When the dipole symmetry group is $U(1)$, the charge is quantized as $q_i \in \mathbb{Z}$, while $q_i \in \mathbb{R}$ when the symmetry group is chosen to be $\mathbb{R}$. When we place the charged object and the symmetry generator inside the spatial slice, the action of the dipole 1-form symmetry is written as

$$
\mathrm{e}^{\mathrm{i}\ell^{-1}\alpha^i Q_i(S)} \mathcal{W}_{\boldsymbol{q}}(C) \mathrm{e}^{-\mathrm{i}\ell^{-1}\alpha^i Q_i(S)} = \mathrm{e}^{\mathrm{i}\boldsymbol{\alpha}\cdot\boldsymbol{q} I(C,S)} \mathcal{W}_{\boldsymbol{q}}(C), \tag{87}
$$

where we used the canonical commutation relation as

$$
\left[ \mathrm{i}\alpha^i Q_i(S), q_j \int_C \mathcal{A}_i \right] = -\left[ \mathrm{i}\alpha^i \int_S (\Pi_i + x_i \pi), q_j \int_C \mathcal{A}_i \right] = \boldsymbol{\alpha}\cdot\boldsymbol{q} \int_C \delta_V(S) = \boldsymbol{\alpha}\cdot\boldsymbol{q}\, I(C,S). \tag{88}
$$

---

[15]Equation (84) should be understood as an operator relation in the path-integral correlation functions,

$$
\langle \mathrm{e}^{\mathrm{i}\alpha Q(S)} W_q(C) \cdots \rangle = \mathrm{e}^{\mathrm{i}\alpha q L(S,C)} \langle W_q(C) \cdots \rangle, \tag{83}
$$

where the dots indicate other operators that are not charged under $Q(S)$ or not linked with $S$.

We can also consider the transformation of the fracton $W_q(C)$ under the dipole 1-form symmetry $Q_i(S)$:

$$e^{i\ell^{-1}\alpha^i Q_i(S)} W_q(C) = \exp\left[i\alpha^i q \frac{x_i}{\ell} L(S,C)\right] W_q(C), \tag{89}$$

where $C$ is placed along the time direction and $x_i$ is its spatial position. We see that the fracton is nontrivially charged under the dipole 1-form symmetry, and the "charge" of $W_q(C)$ under $Q_i(S)$ depends on the spatial position. When we take the dipole symmetry to be $U(1)$, $\alpha_i$ should be $2\pi$-periodic and $x_i/\ell$ should be an integer. For the $\mathbb{R}$ dipole symmetry, such a restriction does not exist.

There are also magnetic symmetries. The conservation law of the magnetic dipole symmetry is given by $\mathrm{d}\mathcal{F}_i = 0$, and this is a 1-form symmetry. In the case of $U(1)$ gauge fields, the field strength $f$ is not closed, and its Bianchi identity can be written as

$$\mathrm{d}(f - x^i \mathcal{F}_i) = 0. \tag{90}$$

The associated charges are the topological operators

$$Q^{\mathrm{m}}(S) := \frac{1}{2\pi}\int_S (f - x^i \mathcal{F}_i), \qquad Q_i^{\mathrm{m}}(S) := \frac{1}{2\pi}\int_S \mathcal{F}_i. \tag{91}$$

The corresponding charged operators are 't Hooft lines, which are regarded as the worldlines of magnetic monopoles,

$$T_p(C) := e^{2\pi i p \int_{S_C} Q(S)}, \qquad \mathcal{T}_{\boldsymbol{p}}(C) := e^{2\pi i p_i \ell^{-1} \int_{S_C} Q_i(S)}, \tag{92}$$

for each charge of Eq. (91). Here, $S_C$ is a two-dimensional surface bounded by $C$, and $p, p_i \in \mathbb{Z}$. Note that dipole-charged magnetic monopoles exist only when the dipole symmetry group is taken to be $U(1)$. If the symmetry group is $\mathbb{R}$, the charge $Q_i^{\mathrm{m}}(S)$ is always trivial, and the dipole-monopoles are absent.

We note that the magnetic symmetry generated by $Q^{\mathrm{m}}(S)$ is also nonuniform. The charges satisfy the algebra

$$[iP_i, Q^{\mathrm{m}}(S)] = \frac{1}{2\pi}\int_S (\partial_i x^j)\mathcal{F}_j = Q_i^{\mathrm{m}}(S). \tag{93}$$

This algebra suggests that the motions of dipole-monopoles are restricted and a monopole charged under $Q_i^{\mathrm{m}}$ cannot move in the $i$-th direction. We will check this using the dual Lagrangian in the following subsection.

## 3.6  Dual theory

The 't Hooft line operators can be seen more explicitly in a dual description. Here we perform a duality transformation to observe their properties. The partition function is $Z = \int \mathcal{D}a\mathcal{D}\mathcal{A}_i \, e^{iS[a,\mathcal{A}_i]}$, where

$$S[a,\mathcal{A}_i] = \int_X \left(-\frac{1}{2e^2} f \wedge \star f - \frac{1}{2(e_1)^2}\mathcal{F}_i \wedge \star \mathcal{F}_i\right). \tag{94}$$

Introducing auxiliary fields, $(\sigma, \tau_i)$, we can write the path integral as $Z = \int \mathcal{D}a\mathcal{D}\mathcal{A}_i\mathcal{D}\sigma\mathcal{D}\tau_i \, e^{iS[a,\mathcal{A}_i,\sigma,\tau_i]}$ with the action

$$S[a,\mathcal{A}_i,\sigma,\tau_i] = \int_X \left[\frac{e^2}{2}\sigma \wedge \star\sigma - \sigma \wedge \star(\mathrm{d}a + \mathcal{A}_i \wedge \mathrm{d}x^i) + \frac{(e_1)^2}{2}\tau_i \wedge \star\tau_i - \tau_i \wedge \star \mathrm{d}\mathcal{A}_i\right]. \tag{95}$$

Solving the equations of motion resulting from varying $\sigma$ and $\tau_i$, we obtain

$$\sigma = \frac{1}{e^2}(\mathrm{d}a + \mathcal{A}_i \wedge \mathrm{d}x_i), \qquad \tau_i = \frac{1}{(e_1)^2}\mathrm{d}\mathcal{A}_i. \tag{96}$$

Integrating out $a$ and $\mathcal{A}_i$ yields the following constraints,

$$\mathrm{d} \star \sigma = 0, \qquad \mathrm{d} \star \tau_i + \mathrm{d}x_i \wedge \star \sigma = 0. \tag{97}$$

These constraints can be solved by taking the variables as

$$\star\sigma = \frac{1}{2\pi}\mathrm{d}\widetilde{a}, \qquad \star\tau_i = \frac{1}{2\pi}\left(\mathrm{d}\widetilde{\mathcal{A}}_i - \widetilde{a} \wedge \mathrm{d}x_i\right). \tag{98}$$

The path integral is thus written in terms of the dual gauge fields, $Z = \int \mathcal{D}\widetilde{a}\mathcal{D}\widetilde{\mathcal{A}}_i\, \mathrm{e}^{iS_{\mathrm{dual}}[\widetilde{a},\widetilde{\mathcal{A}}_i]}$, with the action

$$S_{\mathrm{dual}}[\widetilde{a},\widetilde{\mathcal{A}}_i] = \int_X \left(-\frac{e^2}{2(2\pi)^2}\widetilde{f} \wedge \star\widetilde{f} - \frac{(e_1)^2}{2(2\pi)^2}\widetilde{\mathcal{F}}_i \wedge \star\widetilde{\mathcal{F}}_i\right), \tag{99}$$

where we have introduced the dual field strengths

$$\widetilde{f} := \mathrm{d}\widetilde{a}, \qquad \widetilde{\mathcal{F}}_i := \mathrm{d}\widetilde{\mathcal{A}}_i - \widetilde{a} \wedge \mathrm{d}x_i. \tag{100}$$

Note that the mass dimension of $(\widetilde{\mathcal{A}}_i)_\mu$ is zero for $d = 3$. Their Bianchi identities read

$$\mathrm{d}\widetilde{f} = 0, \qquad \mathrm{d}\widetilde{\mathcal{F}}_i = -\widetilde{f} \wedge \mathrm{d}x_i. \tag{101}$$

The field strengths (100) are invariant under the gauge transformations

$$\delta\widetilde{a} = \mathrm{d}\widetilde{\Lambda}, \qquad \delta\widetilde{\mathcal{A}}_i = \mathrm{d}\widetilde{\Sigma}_i + \widetilde{\Lambda}\mathrm{d}x_i. \tag{102}$$

The 't Hooft line operator for the magnetic $U(1)$ 1-form symmetry can now be written as

$$T_p(C) := \mathrm{e}^{\mathrm{i}p\int_C \widetilde{a}}. \tag{103}$$

This represents the worldline of a charge-monopole. There is no restriction in the choice of $C$. We also have worldlines of dipole-monopoles, given by

$$\mathcal{T}_{\boldsymbol{p}}(C) := \mathrm{e}^{\mathrm{i}p_i\ell^{-1}\int_C \widetilde{\mathcal{A}}_i}. \tag{104}$$

Under the corresponding gauge transformation (102), this transforms as

$$\mathrm{e}^{\mathrm{i}p_i\ell^{-1}\int_C \widetilde{\mathcal{A}}_i} \longmapsto \mathrm{e}^{\mathrm{i}p_i\ell^{-1}\int_C \widetilde{\mathcal{A}}_i}\mathrm{e}^{\mathrm{i}p_i\ell^{-1}\int_C \widetilde{\Lambda}\mathrm{d}x_i}. \tag{105}$$

The operator is gauge-invariant if the tangent vector to $C$ is orthogonal to $\boldsymbol{p}$. Hence, the dipole-monopole $\boldsymbol{p}$ can move in $(d-1)$-directions perpendicular to $\boldsymbol{p}$. In $d = 3$ spatial dimensions they are planons. This is consistent with the expectation based on the underlying algebra (93).

## 3.7 SSB of higher-form symmetries

Here we investigate the stability of the broken-symmetry phase. We refer to Appendix B for additional computational details. The SSB [22, 35] of the 1-form symmetry whose charged objects are $W_q(C)$ can be diagnosed by the existence of an off-diagonal long-range order (ODLRO), which we may identify by studying the behavior of the correlation function

$$\langle W_q(C_1)W_q^\dagger(C_2)\rangle, \tag{106}$$

at large separation of $C_1$ and $C_2$. We take $C_1$ and $C_2$ to be straight lines in the time direction, which is required by gauge invariance. We place them at spatial points $\boldsymbol{x}$ and $\boldsymbol{0}$ and take them to be of finite length $L_t$. We consider the limit $L_t \to \infty$ and $|\boldsymbol{x}| \to \infty$ while keeping $L_t \gg |\boldsymbol{x}|$. The correlation function (106) with minimal charges ($q = 1$) is written as

$$\lim_{\substack{L_t \to \infty \\ |\boldsymbol{x}| \to \infty}} \langle e^{i \int_{C_1} a} e^{-i \int_{C_2} a} \rangle \simeq \lim_{\substack{L_t \to \infty \\ |\boldsymbol{x}| \to \infty}} e^{-\frac{1}{2} \langle \left( \int_{C_1} a - \int_{C_2} a \right)^2 \rangle} . \tag{107}$$

The exponent is written explicitly as

$$\frac{1}{2} \left\langle \left[ \int dt\, a_0(t, \boldsymbol{x}) - \int dt'\, a_0(t', \boldsymbol{0}) \right]^2 \right\rangle = \int dt \int dt' \langle a_0(t, \boldsymbol{x}) a_0(t', \boldsymbol{x}) \rangle$$
$$- \int dt \int dt' \langle a_0(t, \boldsymbol{x}) a_0(t', \boldsymbol{0}) \rangle , \tag{108}$$

where the time integrals are from $-L_t/2$ to $L_t/2$. Choosing the $a_i = 0$ gauge, the Green's function of $a_0$ in the IR is

$$\langle a_0(t, \boldsymbol{x}) a_0(0, \boldsymbol{0}) \rangle = 2(e_1)^2 \int \frac{d\omega}{2\pi} \frac{d^d k}{(2\pi)^d} \frac{e^{-i\omega t + i\boldsymbol{k} \cdot \boldsymbol{x}}}{\boldsymbol{k}^4 + (e_1)^2 \omega^2} . \tag{109}$$

Using this, we can evaluate the $|\boldsymbol{x}|$-dependence of Eq. (108) as

$$\frac{1}{2} \left\langle \left[ \int dx^0 a_0(x^0, \boldsymbol{x}) - \int dy^0 a_0(y^0, \boldsymbol{0}) \right]^2 \right\rangle = (e_1)^2 L_t \int \frac{d^d k}{(2\pi)^d} \frac{1 - e^{i\boldsymbol{k} \cdot \boldsymbol{x}}}{\boldsymbol{k}^4}$$
$$\sim \begin{cases} |\boldsymbol{x}|^{4-d}, & d \neq 4, \\ \ln |\boldsymbol{x}|, & d = 4, \end{cases} \quad \text{at large } |\boldsymbol{x}|. \tag{110}$$

The integral in the exponent is IR-convergent for $d > 4$. Thus, for $d > 4$, by renormalizing the Wilson line operator by a length-dependent counterterm, we can make

$$\lim_{\substack{L_t \to \infty \\ |\boldsymbol{x}| \to \infty}} \langle W_q(C_1) W_q^\dagger(C_2) \rangle = \text{const.}, \tag{111}$$

which indicates that the 1-form symmetry is spontaneously broken.

The IR divergence of Eq. (110) in three spatial dimensions is closely related to the diverging electrostatic energy of fractons discussed in Ref. [6]. Curiously, this does not necessarily imply that the gapless phase is unstable. As discussed in Ref. [5], we cannot write down relevant nor marginally relevant interaction terms that are gauge-invariant. The number of Nambu–Goldstone modes is not affected by the vanishing of Eq. (106), since the dipole 1-form symmetries are still broken in $d = 3$ as we see below, and the gauge field $a$ does not give rise to a physically propagating mode.

Similarly, we can compute the order parameter for the dipole 1-form symmetry,

$$\langle \mathcal{W}_{\boldsymbol{q}}(C_1) \mathcal{W}_{\boldsymbol{q}}^\dagger(C_2) \rangle = \langle e^{i q_i \ell \int_{C_1} \mathcal{A}_i} e^{-i q_j \ell \int_{C_2} \mathcal{A}_j} \rangle \simeq e^{-\frac{1}{2} \ell^2 \langle \left( q_i \int_{C_1} \mathcal{A}_i - q_j \int_{C_2} \mathcal{A}_j \right)^2 \rangle} . \tag{112}$$

The following computation applies to both the cases where the dipole symmetry group is $U(1)$ and $\mathbb{R}$. Let us place $C_1$ and $C_2$ along the time axis at $\boldsymbol{x}$ and $\boldsymbol{0}$, respectively. At low energies, the two-point correlation function of $(\mathcal{A}_i)_0$ can be expressed by that of $a_0$,

$$\ell^2 q_i q_j \langle (\mathcal{A}_i)_0(t, \boldsymbol{x}) (\mathcal{A}_j)_0(0, \boldsymbol{0}) \rangle \simeq \ell^2 q_i q_j \langle \partial_i a_0(t, \boldsymbol{x}) \partial_j a_0(0, \boldsymbol{0}) \rangle$$
$$= 2(e_1)^2 \ell^2 q_i q_j \int \frac{d\omega}{2\pi} \frac{d^d k}{(2\pi)^d} \frac{k_i k_j e^{-i\omega t + i\boldsymbol{k} \cdot \boldsymbol{x}}}{\boldsymbol{k}^4 + (e_1)^2 \omega^2} . \tag{113}$$

Upon integration over $t$ and $t'$,

$$\ell^2 q_i q_j \langle \int \mathrm{d}t \int \mathrm{d}t' (\mathcal{A}_i)_0(t, \boldsymbol{x})(\mathcal{A}_j)_0(t', \boldsymbol{0})\rangle = 2(e_1)^2 \ell^2 L_t q_i q_j \int \frac{\mathrm{d}^d k}{(2\pi)^d} \frac{k_i k_j \mathrm{e}^{\mathrm{i}\boldsymbol{k}\cdot\boldsymbol{x}}}{\boldsymbol{k}^4}$$
$$= 2(e_1)^2 \ell^2 L_t q_i q_j \frac{1}{d}\delta_{ij} \int \frac{\mathrm{d}^d k}{(2\pi)^d} \frac{\mathrm{e}^{\mathrm{i}\boldsymbol{k}\cdot\boldsymbol{x}}}{\boldsymbol{k}^2} \,. \tag{114}$$

The $|\boldsymbol{x}|$-dependence of the exponent of Eq. (112) is evaluated as

$$\frac{\ell^2}{2}\left\langle \left( q_i \int_{C_1} \mathcal{A}_i - q_j \int_{C_2} \mathcal{A}_j \right)^2 \right\rangle = \frac{2}{d}(e_1)^2 \ell^2 L_t \boldsymbol{q}^2 \int \frac{\mathrm{d}^d k}{(2\pi)^d} \frac{1 - \mathrm{e}^{\mathrm{i}\boldsymbol{k}\cdot\boldsymbol{x}}}{\boldsymbol{k}^2}$$
$$\sim \begin{cases} |\boldsymbol{x}|^{2-d}, & d \neq 2, \\ \ln|\boldsymbol{x}|, & d = 2, \end{cases} \quad \text{at large } |\boldsymbol{x}|. \tag{115}$$

This indicates that the order parameter is nonzero for $d > 2$. Compared to the previous case, the IR behavior is milder due to the additional spatial derivatives.

## 3.8 Extended operators at low energies

Here we discuss the low-energy behavior of the line operators. Among the components of the dipole gauge fields, the temporal components $(\mathcal{A}_i)_0$ and the antisymmetric part of the spatial components $(\mathcal{A}_{[i})_{j]}$ are expressed in terms of $a$ and are thus no longer independent degrees of freedom. Let us consider a dipole Wilson line at low energies. If we place the line along the time axis, which we denote by $C_t$,

$$\mathcal{W}_{\boldsymbol{q}}(C_t) = \exp\left[ \mathrm{i}q_i \ell \int_{C_t} (\mathcal{A}_i)_0 \mathrm{d}x^0 \right] \simeq \exp\left[ \mathrm{i}q_i \ell \int_{C_t} e_i \mathrm{d}x^0 \right]. \tag{116}$$

Since $e_i$ is the electric field of the $U(1)$ gauge field $a$, this operator can be interpreted as an electric dipole whose dipole moment is $\boldsymbol{q}\ell$. On a generic loop $C$, the operator $\mathcal{W}_{\boldsymbol{q}}(C)$ in the IR can be written as

$$\mathcal{W}_{\boldsymbol{q}}(C) = \exp\left[ \mathrm{i}q_i \ell \int_C \mathcal{A}_i \right] = \exp\left[ \mathrm{i}q_i \ell \int_C \left( (\mathcal{A}_i)_0 \mathrm{d}x^0 + [(\mathcal{A}_{(i)j)} + (\mathcal{A}_{[i})_{j]}]\mathrm{d}x^j \right) \right]$$
$$= \exp\left[ \mathrm{i}q_i \ell \int_C \left( (\mathcal{A}_i)_0 \mathrm{d}x^0 + [2(\mathcal{A}_{[i})_{j]} + (\mathcal{A}_{(i)j)} - (\mathcal{A}_{[i})_{j]}]\mathrm{d}x^j \right) \right] \tag{117}$$
$$\simeq \exp\left[ \mathrm{i}q_i \ell \int_C \left( \partial_i a - \mathrm{d}a_i + (\mathcal{A}_j)_i \mathrm{d}x^j \right) \right],$$

where we have used Eqs. (62) and (63). This suggests that we can write a line operator in the UV which takes the same form,[16,17]

$$\mathcal{D}_{\boldsymbol{q}}(C) := \exp\left[ \mathrm{i}q_i \ell \int_C \left( \partial_i a - \mathrm{d}a_i + (\mathcal{A}_j)_i \mathrm{d}x^j \right) \right], \tag{121}$$

---

[16]The operator (121) can also be written as

$$\mathcal{D}_{\boldsymbol{q}}(C) = \exp\left[ \mathrm{i}q_i \ell \int_C (i_{e^i} f + \mathcal{A}_i) \right], \tag{118}$$

where $e^i$ is a constant vector field in the $i$-th direction. When written in this form, the gauge invariance of this operator is manifest.

[17]The expression (121) supports our interpretation of this operator as a "dipole" of fractons. To see this, note that the operator can be expressed as

$$\mathcal{D}_{\boldsymbol{q}}(C) = \exp\left[ \mathrm{i}q_i \ell \int_C \left( i_{e^i} \mathrm{d}a + (\mathcal{A}_j)_i \mathrm{d}x^j \right) \right], \tag{119}$$

but is now distinct from $\mathcal{W}_q(C)$ since in the UV, Eqs. (62) and (63) do not hold. Nevertheless, this operator is gauge-invariant along any closed $C$. Under the dipole 1-form symmetry $e^{i\alpha_i Q_i}$,

$$e^{i\ell^{-1}\alpha^i Q_i(S)}\mathcal{D}_q(C)e^{-i\ell^{-1}\alpha^i Q_i(S)} = e^{i\boldsymbol{\alpha}\cdot\boldsymbol{q}I(C,S)}\mathcal{D}_q(C)\,, \tag{122}$$

which transforms in the same way as $\mathcal{W}_q(C)$; see Appendix C for details.[18] This suggests that the two operators can be connected. Indeed, we can also form a composite line operator

$$\exp\left[iq_i\ell\left(\int_{C_-}\mathcal{A}_i + \int_{C_+}\left(\partial_i a - \mathrm{d}a_i + (\mathcal{A}_j)_i\mathrm{d}x^j\right)\right)\right]\,, \tag{123}$$

where $C_+$ and $C_-$ are curves such that $\partial C_+ = -\partial C_-$. This represents a transmutation of the $\mathcal{W}_q$ line operator into the $\mathcal{D}_q$ line operator at a point in spacetime, and is indeed allowed since it is gauge-invariant. See also Sec. 2.1.1 of [41] for a different perspective, based on currents rather than line operators, on the transmutation of a dipole of $U(1)$ charges into a "dipole charge" (object charged under the dipole gauge field).

We can also form additional gauge-invariant line operators from a linear combination of $\mathcal{D}_q(C)$ and $\mathcal{W}_q(C)$,

$$\left(\mathcal{W}_{\alpha q}\mathcal{D}_{\beta q}\right)(C) = \exp\left[iq_i\ell\left(\alpha\int_C\mathcal{A}_i + \beta\int_C\left(\partial_i a - \mathrm{d}a_i + (\mathcal{A}_j)_i\mathrm{d}x^j\right)\right)\right]\,, \tag{124}$$

where $\alpha,\beta$ are arbitrary coefficients. In the IR, $\mathcal{D}_q(C)$, and $\mathcal{W}_q(C)$, and hence all operators of the form Eq. (124) (with $\alpha + \beta = 1$ so that the overall dipole charge is $\boldsymbol{q}$), reduce to the same line operator. In terms of the fields $\{a_0, a_i, A_{ij}\}$ from Sec. 3.3, this is expressed as

$$\exp\left[iq_i\ell\int_C\left(\partial_i a_0\mathrm{d}x^0 + A_{ij}\mathrm{d}x^j - \mathrm{d}a_i\right)\right]\,, \tag{125}$$

where $A_{ij}$ is the symmetric tensor gauge field. This can be understood as the $(3+1)$-dimensional version of a similar operator in $1+1$d from Section 3.1 of [42]. There is also a similar operator in $3+1$d given in Sec. 5.5 of [43], but that operator is a planon that cannot move in the direction of the dipole moment, due to the absence of the diagonal components of the field $A_{ij}$ in the gauge theory. Such kinds of mobility restriction are governed by the algebra presented in Sec. 3.11.

## 3.9 Gauging of nonuniform higher-form symmetries, 't Hooft anomalies and an SPT action

We now introduce background gauge fields for the higher-form symmetries in this theory and discuss their 't Hooft anomalies.[19] Let us introduce notation for the currents of higher-form symmetries, which are not necessarily conserved,

$$\star J^{\mathrm{e}} := \frac{1}{e^2}\star f\,, \qquad \star\mathcal{J}_i^{\mathrm{e}} := \star\mathcal{F}_i\,, \qquad \star J^{\mathrm{m}} := \frac{1}{2\pi}f\,, \qquad \star\mathcal{J}_i^{\mathrm{m}} := \frac{1}{2\pi}\mathcal{F}_i\,. \tag{126}$$

---

since $i_{ei}\mathrm{d}a = \partial_i a - \mathrm{d}a_i$. Recall that in Maxwell theory, a pair of Wilson lines with opposite charges displaced by an infinitesimal distance $\ell$ in the $i$-th direction can be written as

$$\exp\left(iq\ell\int_C\mathcal{L}_{ei}a\right) = \exp\left(iq\ell\int_C i_{ei}\mathrm{d}a\right)\,. \tag{120}$$

[18]When the line operator sits by itself, $\int_C\mathrm{d}a_i$ is a total derivative term and gives no contribution except at the endpoints, but in the presence of a symmetry operator it is no longer trivial and in fact affects the transformation law under the dipole 1-form symmetry.

[19]The analysis here is based on differential forms. In order to capture topologically nontrivial configurations, one needs to use differential characters [44–48], which we do not attempt here.

The equations of motion and Bianchi identities are expressed in terms of these currents as

$$d \star J^{\mathrm{e}} = 0 \,, \tag{127}$$

$$d \star \mathcal{J}_i^{\mathrm{e}} + dx^i \wedge \star J^{\mathrm{e}} = 0 \,, \tag{128}$$

$$d \star J^{\mathrm{m}} - \star \mathcal{J}_i^{\mathrm{m}} \wedge dx^i = 0 \,, \tag{129}$$

$$d \star \mathcal{J}_i^{\mathrm{m}} = 0 \,. \tag{130}$$

We introduce background gauge fields for the currents via

$$S_{\mathrm{cpl}} = \int_X \left( b \wedge \star J^{\mathrm{e}} + B^i \wedge \star \mathcal{J}_i^{\mathrm{e}} + c \wedge \star J^{\mathrm{m}} + C^i \wedge \star \mathcal{J}_i^{\mathrm{m}} \right) , \tag{131}$$

where $b, c, B^i, C^i$ are 2-form gauge fields. To reproduce the conservation laws (127)–(130), their gauge transformation properties should be given by

$$\delta b = d\lambda^{\mathrm{e}} + \Lambda_i^{\mathrm{e}} \wedge dx^i \,, \tag{132}$$

$$\delta B_i = d\Lambda_i^{\mathrm{e}} \,, \tag{133}$$

$$\delta c = d\lambda^{\mathrm{m}} \,, \tag{134}$$

$$\delta C_i = d\Lambda_i^{\mathrm{m}} - \lambda^{\mathrm{m}} \wedge dx^i \,, \tag{135}$$

where $\lambda^{\mathrm{e}}, \lambda^{\mathrm{m}}, \Lambda_i^{\mathrm{e}}, \Lambda_i^{\mathrm{m}}$, are 1-form gauge parameters.

When the dipole symmetry group is $\mathbb{R}$, dipole monopoles do not exist, and the magnetic dipole symmetry is absent. Still, the Bianchi identity (130) is retained, and we can couple the gauge field $C_i$ to the field strength, which corresponds to the insertion of a topological operator, $\int_S \mathcal{F}_i$. As we will see below, in the presence of the background gauge field of the electric dipole 1-form symmetry, this topological property is lost. This is what we mean by the existence of an 't Hooft anomaly in the case of the $\mathbb{R}$ dipole symmetry.

The gauge-invariant 3-form field strengths are given by

$$F_b := db - B_i \wedge dx^i \,, \tag{136}$$

$$F_{B_i} := dB_i \,, \tag{137}$$

$$F_c := dc \,, \tag{138}$$

$$F_{C_i} := dC_i + c \wedge dx^i \,. \tag{139}$$

The field strengths satisfy the Bianchi identities

$$dF_b = -F_{B_i} \wedge dx^i \,, \quad dF_{B_i} = 0 \,, \quad dF_c = 0 \,, \quad dF_{C_i} = F_c \wedge dx^i \,. \tag{140}$$

We can couple the original $U(1)$ charge-dipole theory to background gauge fields $(b, B_i, c, C_i)$ through an action of the form

$$S = \int_X \left[ -\frac{1}{2e^2}(f - b) \wedge \star(f - b) - \frac{1}{2(e_1)^2}(\mathcal{F}_i - B_i) \wedge \star(\mathcal{F}_i - B_i) + c \wedge \star J^{\mathrm{m}} + C_i \wedge \star \mathcal{J}_i^{\mathrm{m}} \right] . \tag{141}$$

For example, invariance under a magnetic gauge transformation by $\lambda^{\mathrm{m}}$ enforces Eq. (129),

$$\begin{aligned} \delta_{\lambda^{\mathrm{m}}} S &= \int_X d\lambda^{\mathrm{m}} \wedge \star J^{\mathrm{m}} - \lambda^{\mathrm{m}} \wedge dx^i \wedge \star \mathcal{J}_i^{\mathrm{m}} \\ &= \int_X \left( \lambda^{\mathrm{m}} \wedge d \star J^{\mathrm{m}} - \lambda^{\mathrm{m}} \wedge dx^i \wedge \star \mathcal{J}_i^{\mathrm{m}} \right) \\ &= \int_X \lambda^{\mathrm{m}} \wedge (d \star J^{\mathrm{m}} - \star \mathcal{J}_i^{\mathrm{m}} \wedge dx^i) \,. \end{aligned} \tag{142}$$

However, the gauged action is not itself gauge invariant, which indicates the presence of an 't Hooft anomaly. The variation of the action under a gauge transformation is[20]

$$\delta S = \frac{1}{2\pi} \int_X \left[ c \wedge (d\lambda^e + \Lambda_i^e \wedge dx^i) + C_i \wedge d\Lambda_i^e \right].$$
(145)

Thus, the partition function of this system is not invariant under a gauge transformation but is changed by a $U(1)$ phase,

$$Z_X[b + \delta b, B_i + \delta B_i, c + \delta c, C_i + \delta C_i] = \exp\left[ \frac{i}{2\pi} \int_X \left[ c \wedge (d\lambda^e + \Lambda_i^e \wedge dx^i) + C_i \wedge d\Lambda_i^e \right] \right]$$
$$\times Z_X[b, B_i, c, C_i].$$
(146)

This 't Hooft anomaly can be matched by the following bulk SPT action in five dimensions,[21]

$$S_{\text{bulk}} = -\frac{1}{2\pi} \int_Y \left[ dc \wedge b + (dC^i + c \wedge dx^i) \wedge B_i \right],$$
(147)

where $Y$ is a five-dimensional manifold whose boundary is $X$, $\partial Y = X$. Note that the SPT action is gauge-invariant when placed on a closed five-dimensional manifold. The combined partition function of the bulk and boundary systems,

$$Z_X[b, B_i, c, C_i] e^{iS_{\text{bulk}}},$$
(148)

is gauge invariant. The corresponding six-dimensional anomaly polynomial is

$$I^{(6)} = \frac{1}{4\pi^2} \left( F_c \wedge F_b + F_{C_i} \wedge F_{B_i} \right).$$
(149)

The anomaly polynomial is manifestly gauge-invariant. Although each term is not closed, the combination of the two terms is closed,

$$dI^{(6)} = \frac{1}{4\pi^2} \left( -F_C \wedge dF_b + dF_{C_i} \wedge F_{B_i} \right) = \frac{1}{4\pi^2} \left( F_c \wedge F_{B_i} \wedge dx^i + F_c \wedge dx^i \wedge F_{B_i} \right) = 0.$$
(150)

This can also be seen from the fact that the anomaly polynomial is the exterior derivative of the SPT Lagrangian,

$$d\mathcal{L}_{\text{bulk}} = -\frac{1}{2\pi} \left( F_c \wedge db + F_c \wedge dx^i \wedge B_i - F_{C_i} \wedge F_{B_i} \right) = \frac{1}{2\pi} \left( F_c \wedge F_b + F_{C_i} \wedge F_{B_i} \right) = 2\pi I^{(6)}.$$
(151)

## 3.10 Higgsing and fracton order

Here we illustrate that the Higgsing of the pair of gauge fields $(a, \mathcal{A}_i)$ leads to a theory with fractonic order.[22] To this end, we introduce matter fields denoted by $\theta$ and $\phi_i$, which are

---

[20]Note that the magnetic currents are transformed nontrivially under electric gauge transformations as

$$\star J^m \mapsto \star J^m + \frac{1}{2\pi} \left( d\lambda^e + \Lambda_i^e \wedge dx^i \right),$$
(143)

$$\star \mathcal{J}_i^m \mapsto \star \mathcal{J}_i^m + \frac{1}{2\pi} d\Lambda_i^e.$$
(144)

[21]Note that the combination $dC^i + c \wedge dx^i$ is invariant under magnetic gauge transformations.

[22] We can also consider the coupling of gauge fields to matter fields in a non-Higgsed phase. In Appendix A, we describe the coupling of gauge fields to a model with a complex scalar field.

charged under the $U(1)$ charge symmetry and $U(1)$ dipole symmetry, respectively. The coupling of the gauge fields $a, \mathcal{A}_i$ to matter with charge $n$ can be introduced through covariant derivatives of the form

$$D\theta := \mathrm{d}\theta + na + \phi_i \mathrm{d}x^i, \qquad D\phi_i := \mathrm{d}\phi_i + n\mathcal{A}_i. \tag{152}$$

The corresponding Lagrangian reads

$$\mathcal{L}[\theta, \phi_i, a, \mathcal{A}_i] = -\frac{v}{2}|\mathrm{d}\theta + na + \phi_i \mathrm{d}x^i|^2 - \frac{w}{2}|\mathrm{d}\phi_i + n\mathcal{A}_i|^2 + (\text{kinetic terms for } a \text{ and } \mathcal{A}_i), \tag{153}$$

where $|\omega|^2 := \omega \wedge \star\omega$.

Since all the gauge fields are now gapped, we can drop the kinetic terms of $a$ and $\mathcal{A}_i$. At low energies, the covariant derivative should vanish, and we can write the action as

$$\mathcal{L}[h, H_i, \theta, \phi_i, a, \mathcal{A}_i] = h \wedge (\mathrm{d}\theta + na + \phi_i \mathrm{d}x^i) + H_i \wedge (\mathrm{d}\phi_i + n\mathcal{A}_i), \tag{154}$$

where $h$ and $H_i$ are auxiliary 3-form fields. By integrating out $\theta$, we obtain the constraint $\mathrm{d}h = 0$, which can be explicitly solved by taking $h = \frac{1}{2\pi}\mathrm{d}b$. Integration over $\phi_i$ also gives a constraint

$$\mathrm{d}H_i + \frac{1}{2\pi}\mathrm{d}b \wedge \mathrm{d}x_i = 0. \tag{155}$$

This can be solved by $H_i = \frac{1}{2\pi}(\mathrm{d}C_i + b \wedge \mathrm{d}x_i)$ where $C_i$ is a 2-form gauge field. The resulting Lagrangian is

$$\mathcal{L}[a, \mathcal{A}_i, b, C_i] = \frac{n}{2\pi}\mathrm{d}b \wedge a + \frac{n}{2\pi}(\mathrm{d}C_i + b \wedge \mathrm{d}x_i) \wedge \mathcal{A}_i. \tag{156}$$

If we further restrict the configuration of $C_i$ to the form $\mathrm{d}C_i = \mathrm{d}B_i \wedge \mathrm{d}x_i$ (no summation over $i$), we reproduce the Lagrangian of the foliated field theory [41, 49], which is a low-energy theory of the X-cube model [50]. Originally, the current $k_i$ that couples to $\mathcal{A}_i$ is unconstrained aside from $\mathrm{d}\star k_i = -\mathrm{d}x_i \wedge \star j$, which is implemented through gauge-invariance under $\mathcal{A}_i \mapsto \mathcal{A}_i + \mathrm{d}\Sigma_i$ and $a \mapsto a - \Sigma_i \mathrm{d}x_i$. If we make the choice $\mathrm{d}C_i = \mathrm{d}B_i \wedge \mathrm{d}x_i$, there is an additional gauge transformation $\mathcal{A}_i \mapsto \mathcal{A}_i + f \mathrm{d}x_i$ where $f$ is an arbitrary function. Imposing invariance under this transformation leads to

$$\mathrm{d}x_i \wedge \star k_i = 0, \tag{157}$$

which constrains $k_i$ to have no component along the $i$-th direction, i.e. it is constrained to move in a plane normal to the $i$-th axis. This can also be seen from the dipole Wilson lines: $\mathrm{e}^{\mathrm{i}\ell \int_C \mathcal{A}_i}$ is only invariant under $\mathcal{A}_i \mapsto \mathcal{A}_i + f \mathrm{d}x_i$ if the curve $C$ has no component along the $i$-th direction.

## 3.11 Variants of the scalar charge gauge theory

Let us briefly show that two variants of the scalar charge gauge theory can also be obtained by employing a slightly modified algebra. One is the traceless scalar charge gauge theory [6], and the other is a theory where the gauge fields have only off-diagonal components [43, 51–53].

We first discuss the traceless scalar charge gauge theory. In addition to $Q$ and $Q_i$, we introduce another charge $Q'$ such that

$$[\mathrm{i}P_i, Q'] = Q_i, \tag{158}$$

while keeping all other commutation relations the same as before. This algebra implies that the dipoles in the gauge theory will become immobile in the direction of the dipole charge vector. To match this algebra, the current of the charge $Q'(V) = \int_V \star K'$ should be of the form

$$\star K' = \star k' - \frac{1}{2}x^2 \star j - x_i \star k_i. \tag{159}$$

One can check explicitly that this indeed reproduces Eq. (158),

$$[iP_i, Q'] = \int_V \left( \frac{1}{2} (\partial_i \boldsymbol{x}^2) \star j + \delta_{ij} \star k_j \right) = \int_V (\star k_i + x_i \star j) = Q_i \,. \tag{160}$$

The conservation law of the current $K'$ is written as

$$\mathrm{d} \star k' - \mathrm{d}x_i \wedge \star k_i = 0 \,, \tag{161}$$

where we have used the conservation laws (37) and (40) for $j$ and $k_i$, respectively. A coupling to gauge fields can be introduced via

$$S_{\mathrm{cpl}} = \int_X \left( a \wedge \star j + \mathcal{A}_i \wedge \star k_i + a' \wedge \star j' \right) \,. \tag{162}$$

The conservation laws are implemented by demanding invariance under the gauge transformation

$$\delta a = \mathrm{d}\Lambda - \Sigma_i \mathrm{d}x^i \,, \qquad \delta \mathcal{A}_i = \mathrm{d}\Sigma_i + \Lambda' \mathrm{d}x_i \,, \qquad \delta a' = \mathrm{d}\Lambda' \,. \tag{163}$$

Note that the dipole gauge field $\mathcal{A}_i$ is transformed by the gauge parameter for $Q'$. As a result, the path of the Wilson line of $\mathcal{A}_i$ should be placed within a constant-$x_i$ plane, meaning that a dipole cannot move in the direction of its dipole moment, as expected from the algebra (158). The gauge-invariant field strength for the dipole gauge field is

$$\mathcal{F}_i := \mathrm{d}\mathcal{A}_i - a' \wedge \mathrm{d}x_i \,. \tag{164}$$

If we write a quadratic Lagrangian as in previous examples, the gauge field $a'$ obtains a mass term. As a result, we have the condition $\mathrm{d}x_i \wedge \mathcal{F}_i = 0$ at low energies. This gives a constraint on the dipole electric field,

$$(\mathcal{E}_i)_i = -3a'_0 \,. \tag{165}$$

Note that the trace of the dipole electric field is $Q'$-gauge variant and it can be set to zero, which means that the electric field tensor at low energies is traceless. In this way, we obtain the traceless scalar charge theory in the IR from the modified algebra.

Finally, let us show how to obtain a theory whose low-energy limit reduces to a rank-2 symmetric gauge theory whose gauge fields have only off-diagonal components [43, 51–54]. For the rest of this subsection, we will explicitly write the summation symbol in order to avoid confusion. To eliminate the diagonal components, we introduce a new charge $q_j$ in addition to the charges $Q$ and $Q_i$, such that

$$[iP_i, q_j] = \delta_{ij} Q_j \,, \tag{166}$$

is satisfied. From this algebra, we can expect that a dipole cannot move in the direction of its dipole moment, i.e. a dipole is a planon. To satisfy the algebra (166), the conserved current of $q_i$ should take the form

$$\star j_i = \star j'_i + \frac{1}{2} (x_i)^2 \star j - x_i \star k_i \,. \tag{167}$$

The conservation law of $q_i$ is given by

$$\mathrm{d} \star j_i = \mathrm{d} \star j'_i - \mathrm{d}x_i \wedge \star k_i = 0 \,, \tag{168}$$

where we have used Eqs. (37) and (40). We then introduce the coupling to the gauge fields by

$$S_{\mathrm{cpl}} = \int_X \left( a \wedge \star j + \sum_i \mathcal{A}_i \wedge \star k_i + \sum_i a'_i \wedge \star j'_i \right) \,. \tag{169}$$

To reproduce the conservation law (168), the dipole gauge field should transform as

$$\delta \mathcal{A}_i = \mathrm{d}\Sigma_i + \lambda'_i \mathrm{d}x_i \,, \tag{170}$$

where $\lambda'_i$ is the gauge parameter for $q_i$. Explicitly, the transformation of the spatial components is

$$\sum_j (\mathcal{A}_i)_j \mathrm{d}x^j \mapsto \sum_j ((\mathcal{A}_i)_j + \delta_{ij}\lambda'_i)\mathrm{d}x^j \,. \tag{171}$$

Namely, the gauge transformation corresponding to the charge $q_i$ shifts the diagonal components of $(\mathcal{A}_i)_j$. By using the gauge degrees of freedom, we can take the diagonal components of $(\mathcal{A}_i)_j$ to be zero, and we are left with a gauge field with only off-diagonal components.

## 4 Vector charge gauge theory from nonuniform symmetries

In this section, we discuss a gauge theory that hosts lineons. The gauge theory reduces to the vector charge gauge theory at low energies. We first discuss the case of $d = 3$, and later generalize the theory to arbitrary spatial dimensions.

### 4.1 Algebra, gauging, and an effective Lagrangian

We consider a set of generators, $\{Q_i, q_i\}_{i=1\ldots d}$ with $d = 3$, that commute with each other and have the following commutation relations with spatial translations:

$$[iP_i, Q_j] = \epsilon_{ijk}q_k \,, \qquad [iP_i, q_j] = 0 \,. \tag{172}$$

Each $q_i$ can be written as a commutator of a translation and another charge, for example,

$$q_z = [iP_x, Q_y] = [iP_y, -Q_x] \,. \tag{173}$$

Because of this, a particle with charge $q_z$ is immobile in the $x$ and $y$ directions. Thus, a particle characterized by a charge vector $\boldsymbol{q}$ can only move along $\boldsymbol{q}$, which means that it is a lineon. By gauging the symmetries corresponding to $q_i$ and $Q_i$, we can construct a gauge theory that implements these mobility restrictions.

The current 1-forms of $q_i$ and $Q_i$ are denoted by $j_i$ and $K_i$, respectively, and the charges are written as

$$q_i = \int_V \star j_i \,, \qquad Q_i = \int_V \star K_i \,, \tag{174}$$

where $V$ is a 3-cycle. Consistency with the algebra (172) requires that the current $K_i$ be of the form

$$\star K_i = \star J_i + \epsilon_{ijk}x_j \star j_k \,, \tag{175}$$

where $J_i$ and $j_k$ are uniform. One can check that this indeed reproduces the algebra,

$$[iP_i, Q_j] = -\int_V \epsilon_{jkl}(\partial_i x_k) \star j_l = \epsilon_{ijl}q_l \,. \tag{176}$$

The conservation laws are written as

$$\mathrm{d}\star j_i = 0 \,, \qquad \mathrm{d}\star J_i = -\epsilon_{ijk}\mathrm{d}x_j \wedge \star j_k \,. \tag{177}$$

We introduce a coupling to gauge fields via the action

$$S_{\mathrm{cpl}} = \int_X (a_i \wedge \star j_i + \mathcal{A}_i \wedge \star J_i) \,. \tag{178}$$

This is required to be invariant under the gauge transformations

$$\delta a_i = \mathrm{d}\sigma_i - \epsilon_{ijk}\Sigma_j \mathrm{d}x_k\,, \qquad \delta\mathcal{A}_i = \mathrm{d}\Sigma_i\,. \tag{179}$$

Gauge-invariant field strengths are then defined by

$$f_i := \mathrm{d}a_i + \epsilon_{ijk}\mathcal{A}_j \wedge \mathrm{d}x_k\,, \qquad \mathcal{F}_i := \mathrm{d}\mathcal{A}_i\,. \tag{180}$$

Similarly to the theory discussed in Sec. 3, we can choose the symmetry group to be either $U(1)$ or $\mathbb{R}$. We choose the symmetry of each charge $q_i$ to be a $U(1)$ symmetry, and we normalize the gauge fields and gauge transformation parameters as

$$\int_S \mathrm{d}a_i \in 2\pi\mathbb{Z}\,, \qquad \int_C \mathrm{d}\sigma_i \in 2\pi\mathbb{Z}\,. \tag{181}$$

Meanwhile, taking the symmetries of the $Q_i$ also as $U(1)$ implies the normalization

$$\ell \int_S \mathcal{F}_i \in 2\pi\mathbb{Z}\,, \qquad \ell \int_C \mathrm{d}\Sigma_i \in 2\pi\mathbb{Z}\,, \tag{182}$$

where $\ell$ is a parameter with the dimension of length. If we instead choose the symmetry group to be $\mathbb{R}$, the integrals in Eq. (182) are zero.

For this theory, we consider the effective action

$$S[a_i, \mathcal{A}_i] = \int_X \left( -\frac{1}{2e^2} f_i \wedge \star f_i - \frac{1}{2v^2} \mathcal{F}_i \wedge \star \mathcal{F}_i \right)\,. \tag{183}$$

The equations of motion and Bianchi identities are then written as

$$-\frac{1}{e^2} \mathrm{d} \star f_i = 0\,, \tag{184}$$

$$-\frac{1}{v^2} \mathrm{d} \star \mathcal{F}_i - \frac{1}{e^2} \epsilon_{ijk} \mathrm{d}x_j \wedge \star f_k = 0\,, \tag{185}$$

$$\mathrm{d}f_i - \epsilon_{ijk}\mathcal{F}_j \wedge \mathrm{d}x_k = 0\,, \tag{186}$$

$$\mathrm{d}\mathcal{F}_i = 0\,. \tag{187}$$

## 4.2 Low energy limit and the relation to the vector charge gauge theory

The gauge field $\mathcal{A}_i$ has a mass term due to the structure of the invariant field strength (180). In the low-energy limit, $m^2 := v^2/e^2 \to \infty$, the equation of motion (185) reduces to

$$\epsilon_{ijk} \mathrm{d}x_j \wedge \star f_k = 0\,. \tag{188}$$

Expanding $\star f_i$ in components as a spacetime 2-form yields the constraints

$$(\star f_i)_{j0} = 0\,, \qquad (\star f_i)_{ji} = 0\,. \tag{189}$$

Note that the second expression involves a sum over $i$: in the vector theory, not all components of $f_i$ vanish in the low-energy limit. We may write $f_i$ explicitly as

$$f_i = \big[(e_i)_k - \epsilon_{ijk}(\mathcal{A}_j)_0\big]\mathrm{d}x^0 \wedge \mathrm{d}x^k + \big[\partial_{[j}(a_i)_{k]} - \epsilon_{il[j}(\mathcal{A}_{l)k]}\big]\mathrm{d}x^j \wedge \mathrm{d}x^k\,, \tag{190}$$

where $(e_i)_k := \partial_k(a_i)_0 - \partial_0(a_i)_k$ is the electric field of $\mathrm{d}a_i$ (note that this is not gauge-invariant). Taking the Hodge dual and matching to (189), we obtain

$$\epsilon_{ijk}(\mathcal{A}_j)_0 = (e_{[i})_{k]}\,, \tag{191}$$

$$\epsilon_{il[j}(\mathcal{A}_{l)k]} = \partial_{[j}(a_i)_{k]}\,, \tag{192}$$

where in the second equation the skew-symmetrization $[\dots]$ applies to $j$ and $k$ only.[23] Contracting Eq. (192) with the $\epsilon$ tensor gives

$$\frac{1}{2}\left[(\mathcal{A}_m)_k + \delta_{mk}(\mathcal{A}_l)_l\right] = \epsilon_{mij}\partial_{[i}(a_j)_{k]}. \tag{193}$$

By taking the trace of this, we find

$$(\mathcal{A}_i)_i = \frac{1}{2}\epsilon_{ijk}\partial_i(a_j)_k. \tag{194}$$

Thus, we can express the components of $\mathcal{A}_i$ as

$$(\mathcal{A}_i)_0 = -\frac{1}{2}\epsilon_{ijk}(e_j)_k, \quad (\mathcal{A}_m)_k = 2\left[\epsilon_{mij}\partial_{[i}(a_j)_{k]} - \frac{1}{4}\delta_{mk}\,\epsilon_{ijl}\,\partial_i(a_j)_l\right]. \tag{195}$$

We find that all components of $\mathcal{A}_i$ are expressed in terms of $a_i$. Since the spatial components of $\mathcal{A}_i$ are written as a derivative of the spatial components of $a_i$, the dispersion relation becomes quadratic, $\omega \sim k^2$.

Let us show explicitly that this theory reduces to the vector charge gauge theory. To do this, it is convenient to introduce electric and magnetic fields for $f_i$,

$$f_i = (\widetilde{e}_i)_k \mathrm{d}x^k \wedge \mathrm{d}x^0 + \frac{1}{2}\epsilon_{jkl}(\widetilde{b}_i)_l \mathrm{d}x^j \wedge \mathrm{d}x^k, \tag{196}$$

where

$$(\widetilde{e}_i)_k := (e_i)_k - \epsilon_{ijk}(\mathcal{A}_j)_0, \tag{197}$$

$$(\widetilde{b}_i)_m := \epsilon_{mkj}\left[\partial_k(a_{(i)j)} - \epsilon_{ilk}(\mathcal{A}_l)_j\right] = (b_i)_m - (\mathcal{A}_m)_i + \delta_{mi}(\mathcal{A}_l)_l. \tag{198}$$

We use the gauge transformation of $\mathcal{A}_i$ to eliminate the antisymmetric part $(a_{[i})_{j]}$. The symmetric part will be denoted as $a_{ij} := (a_{(i})_{j)}$. The electric and magnetic fields for $a_i$ are written as

$$(e_i)_k = \partial_k(a_i)_0 - \partial_0 a_{ik}, \qquad (b_i)_m = \epsilon_{mkj}\partial_k a_{ij}. \tag{199}$$

Note that $(b_i)_m$ is traceless, $(b_i)_i = 0$. The Lagrangian density is

$$\begin{aligned}
\mathcal{L} &= \frac{1}{2e^2}(\widetilde{e}_i)_k(\widetilde{e}_i)_k - \frac{1}{2e^2}(\widetilde{b}_i)_k(\widetilde{b}_i)_k + \frac{1}{2v^2}(\mathcal{E}_i)_k(\mathcal{E}_i)_k - \frac{1}{2v^2}(\mathcal{B}_i)_k(\mathcal{B}_i)_k \\
&= \frac{1}{2e^2}(e_{(i)k})(e_{(i)k}) + \frac{1}{2e^2}\left[(e_{[i})_{k]} - \epsilon_{ijk}(\mathcal{A}_j)_0\right]\left[(e_{[i})_{k]} - \epsilon_{ij'k}(\mathcal{A}_{j'})_0\right] \\
&\quad - \frac{1}{2e^2}(\widetilde{b}_i)_k(\widetilde{b}_i)_k + \frac{1}{2v^2}(\mathcal{E}_i)_k(\mathcal{E}_i)_k - \frac{1}{2v^2}(\mathcal{B}_i)_k(\mathcal{B}_i)_k,
\end{aligned} \tag{200}$$

where $(\mathcal{E}_i)_k = \partial_k(\mathcal{A}_i)_0 - \partial_0(\mathcal{A}_i)_k$ and $(\mathcal{B}_l)_i = \epsilon_{ijk}\partial_j(\mathcal{A}_l)_k$ are the electric and magnetic fields for $\mathcal{A}_i$. The gauge fields $\mathcal{A}_i$ have mass terms and can be integrated out at low energies. The condition $(\widetilde{b}_i)_k = 0$ can be solved for $(\mathcal{A}_m)_i$ as

$$(\mathcal{A}_m)_i = (b_i)_m - \frac{1}{d-1}\delta_{mi}(b_l)_l. \tag{201}$$

The second term vanishes in the present gauge. The electric and magnetic fields for $\mathcal{A}_i$ are now given by

$$(\mathcal{E}_i)_k = \frac{1}{2}\epsilon_{ijl}\partial_k(e_j)_l - \partial_0(b_k)_i, \qquad (\mathcal{B}_l)_i = \epsilon_{ijk}\epsilon_{lmn}\partial_j\partial_m a_{kn}. \tag{202}$$

---

[23]We employ the convention that (skew-)symmetrization does not mix spacetime component indices with field label indices, unless explicitly indicated. Thus, for example, $(e_{[i})_{k]} = \frac{1}{2}((e_i)_k - (e_k)_i)$, whereas $\partial_{[j}(a_i)_{k]} = \frac{1}{2}(\partial_j(a_i)_k - \partial_k(a_i)_j)$.

After the integration of gapped modes, the Lagrangian is written as

$$\mathcal{L} = \frac{1}{2e^2}(e_{(i)k})(e_{(i)k}) + \frac{1}{2v^2}(\mathcal{E}_i)_k(\mathcal{E}_i)_k - \frac{1}{2v^2}(\mathcal{B}_i)_k(\mathcal{B}_i)_k. \tag{203}$$

The system has a quadratic dispersion relation, $\omega \sim k^2$, and the second term is of higher order, so we drop it. We define a symmetric electric field by

$$e_{ij} := (e_{(i)j)} = \partial_{(j}(a_{i)})_0 - \partial_0 a_{ij} = -\partial_0 a_{ij} - \partial_{(j}\phi_{i)}, \tag{204}$$

where we have introduced scalar potentials $\phi_i := -(a_i)_0$. The resulting Lagrangian is

$$\mathcal{L} = \frac{1}{2e^2}e_{ij}e_{ij} - \frac{1}{2v^2}(\mathcal{B}_i)_j(\mathcal{B}_i)_j. \tag{205}$$

This is the Lagrangian density of the vector charge gauge theory.

Similarly to the case of the scalar charge gauge theory discussed in Sec. 3.3, the equations of motion of the vector charge gauge theory may be recovered from Eqs. (184)–(187). In this case, Eq. (185) tells us that the non-vanishing contributions to $(\widetilde{e}_{[i})_{j]}$ and $(\widetilde{b}_i)_j$ arise at order $e^2/v^2$, c.f. Eqs. (191) and (192). It then follows from Eq. (186) that $(\mathcal{B}_{[i})_{j]}$ is also $\mathcal{O}(e^2/v^2)$, implying that in the low-energy limit, $(\widetilde{e}_i)_j$ and $(\mathcal{B}_i)_j$ are both symmetric. Meanwhile, the other component of Eq. (186) yields a relation between $(\mathcal{E}_i)_j$ and the other modes, which in the low-energy limit takes the simple form

$$(\mathcal{E}_i)_j = \epsilon_{ikl}\partial_k(\widetilde{e}_i)_l. \tag{206}$$

Applying Eq. (206) and the other low-energy constraints to Eqs. (184) and (187), we obtain the full set of equations of motion of the vector charge gauge theory,

$$\partial_j e_{ij} = 0, \tag{207}$$

$$\partial_0 e_{ij} - \frac{e^2}{v^2}\epsilon_{ikm}\epsilon_{jln}\partial_k\partial_l(\mathcal{B}_m)_n = 0, \tag{208}$$

$$\partial_0(\mathcal{B}_i)_j + \epsilon_{ikm}\epsilon_{jln}\partial_k\partial_l e_{mn} = 0, \tag{209}$$

$$\partial_j(\mathcal{B}_i)_j = 0. \tag{210}$$

Note that to obtain Eq. (208) we have included contributions up to $\mathcal{O}(e^2/v^2)$ and dropped terms with three or more derivatives, which give a sub-leading correction to the dispersion relation.

## 4.3 Extended operators and higher-form symmetries

The generators of electric 1-form symmetries are

$$q_i(S) := \int_S \frac{1}{e^2} \star f_i, \qquad Q_i(S) := \int_S \left(\frac{1}{v^2} \star \mathcal{F}_i + \frac{1}{e^2}\epsilon_{ijk}x_j \star f_k\right). \tag{211}$$

The corresponding Wilson line operators are

$$w_{\boldsymbol{q}}(C) := e^{iq_i \int_C a_i}, \qquad \mathcal{W}_{\boldsymbol{q}}(C) := e^{iq_i\ell \int_C \mathcal{A}_i}, \tag{212}$$

where $q_i \in 2\pi\mathbb{Z}$ for $w_{\boldsymbol{q}}(C)$. For $\mathcal{W}_{\boldsymbol{q}}(C)$, the charge $q_i \in \mathbb{Z}$ if the symmetry group of $Q_i$ is chosen to be $U(1)$, and $q_i \in \mathbb{R}$ when we take the symmetry to be $\mathbb{R}$. The fractonic property of $w_{\boldsymbol{q}}(C)$ can be seen by noting that the exponent of $w_{\boldsymbol{q}}(C)$ is transformed under a $\Sigma_i$-gauge transformation as

$$\delta\left(q_i \int_C a_i\right) = -q_i \int_C \epsilon_{ijk}\Sigma_j \frac{dx_k}{ds}dx. \tag{213}$$

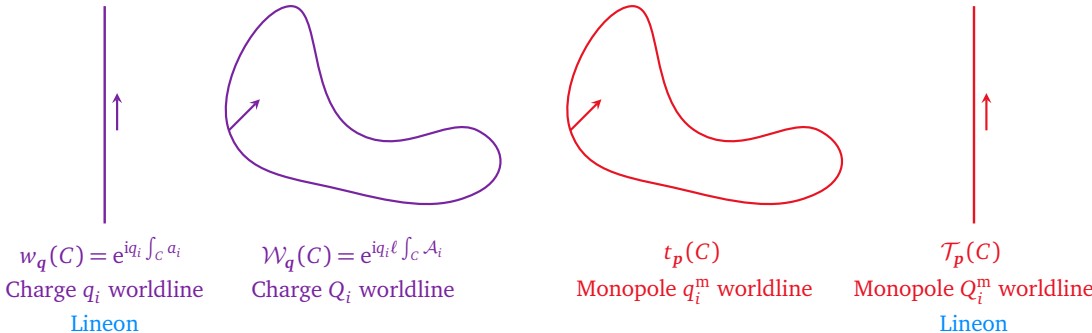

$$w_q(C) = e^{iq_i \int_C a_i}$$
Charge $q_i$ worldline
Lineon

$$\mathcal{W}_q(C) = e^{iq_i \ell \int_C \mathcal{A}_i}$$
Charge $Q_i$ worldline

$$t_p(C)$$
Monopole $q_i^{\mathrm{m}}$ worldline

$$\mathcal{T}_p(C)$$
Monopole $Q_i^{\mathrm{m}}$ worldline
Lineon

Figure 3: Summary of charged objects in this theory. The operators $w_q(C)$ and $\mathcal{T}_p(C)$ are lineons and they have to be placed in the direction of their charge vectors or in the time direction.

This is invariant only when the tangent vector of $C$ is either along $\boldsymbol{q}$ or is in the time direction. Thus, a particle with charge $\boldsymbol{q}$ can move only in the direction of $\boldsymbol{q}$ and it is a lineon. This is consistent with the general expectation from the algebra (172). For the Wilson lines of $\mathcal{A}_i$, $\mathcal{W}_q(C)$, there is no restriction on $C$ and they do not have any mobility constraint.

The generators of the corresponding magnetic 1-form symmetries are

$$q_i^{\mathrm{m}}(S) := \frac{1}{2\pi} \int_S (f_i + \epsilon_{ijk} x_j \mathcal{F}_k), \qquad Q_i^{\mathrm{m}}(S) := \frac{1}{2\pi} \int_S \mathcal{F}_i. \tag{214}$$

The charged operators are the 't Hooft line operators, which can be written as

$$t_p(C) := e^{ip_i q_i(S_C)}, \qquad \mathcal{T}_p(C) := e^{ip_i \ell^{-1} Q_i(S_C)}, \tag{215}$$

where $S_C$ is a two-dimensional surface bounded by $C$. When the symmetry group for $Q_i$ is taken to be $\mathbb{R}$, the monopole $\mathcal{T}_p(C)$ does not exist. The charges of magnetic 1-form symmetries satisfy the algebra

$$[iP_i, q_j^{\mathrm{m}}(S)] = \epsilon_{ijk} Q_k^{\mathrm{m}}(S), \qquad [iP_i, Q_j^{\mathrm{m}}(S)] = 0. \tag{216}$$

From this we can see that monopoles charged under $Q_i^{\mathrm{m}}$ have restricted mobility. The operator $\mathcal{T}_p(C)$ can be placed in such a way that the tangent vector to $C$ is along $\boldsymbol{p}$ (or in the time direction). Thus, these monopoles are lineons.

## 4.4 't Hooft anomaly and SPT action

Analogously to the case of $U(1)$ and dipole gauge theory discussed in Sec. 3.9, we can gauge the higher-form symmetries of the vector charge theory by introducing background gauge fields to detect 't Hooft anomalies.[24] In the present case, we can define currents for the higher-form symmetries as

$$\star J_i^{\mathrm{e}} := \frac{1}{e^2} \star f_i, \qquad \star \mathcal{J}_i^{\mathrm{e}} := \frac{1}{v^2} \star \mathcal{F}_i, \qquad \star J_i^{\mathrm{m}} := \frac{1}{2\pi} f_i, \qquad \star \mathcal{J}_i^{\mathrm{m}} := \frac{1}{2\pi} \mathcal{F}_i. \tag{217}$$

---

[24]On the choice of $U(1)$ or $\mathbb{R}$ for the symmetry $Q_i$, the same comment applies as Sec. 3.9: when the symmetry group of $Q_i$ is $\mathbb{R}$, there is no charged object under this symmetry, but we can insert the magnetic 1-form current operator in the path-integral. An "'t Hooft anomaly" in this case means that the topological property of this operator is lost.

The equations of motion and Bianchi identities can thus be written as

$$d \star J_i^{\mathrm{e}} = 0 \,, \tag{218}$$

$$d \star \mathcal{J}_i^{\mathrm{e}} + \epsilon_{ijk} dx^j \wedge \star J_k^{\mathrm{e}} = 0 \,, \tag{219}$$

$$d \star J_i^{\mathrm{m}} - \epsilon_{ijk} \star \mathcal{J}_j^{\mathrm{m}} \wedge dx^k = 0 \,, \tag{220}$$

$$d \star \mathcal{J}_i^{\mathrm{m}} = 0 \,. \tag{221}$$

These currents can be coupled to background gauge fields $b_i, B_i, c_i, \mathcal{C}_i$ via the action

$$S_{\mathrm{cpl}} = \int_X \left( b_i \wedge \star J_i^{\mathrm{e}} + B^i \wedge \star \mathcal{J}_i^{\mathrm{e}} + c_i \wedge \star J_i^{\mathrm{m}} + \mathcal{C}^i \wedge \star \mathcal{J}_i^{\mathrm{m}} \right) \,. \tag{222}$$

With the gauge transformations

$$\delta b_i = d\lambda_i^{\mathrm{e}} + \epsilon_{ijk} \Lambda_j^{\mathrm{e}} \wedge dx^k \,, \tag{223}$$

$$\delta B_i = d\Lambda_i^{\mathrm{e}} \,, \tag{224}$$

$$\delta c_i = d\lambda_i^{\mathrm{m}} \,, \tag{225}$$

$$\delta \mathcal{C}_i = d\Lambda_i^{\mathrm{m}} + \epsilon_{ijk} \lambda_j^{\mathrm{m}} \wedge dx^k \,, \tag{226}$$

gauge invariance of $S_{\mathrm{cpl}}$ reproduces the higher-form current conservation equations, i.e. the original equations of motion and Bianchi identities (218)–(221). The corresponding field strengths are

$$F_{b_i} := db_i - \epsilon_{ijk} B_j \wedge dx^k \,, \tag{227}$$

$$F_{B_i} := dB_i \,, \tag{228}$$

$$F_{c_i} := dc_i \,, \tag{229}$$

$$F_{\mathcal{C}_i} := d\mathcal{C}_i - \epsilon_{ijk} c_j \wedge dx^k \,, \tag{230}$$

which satisfy the Bianchi identities

$$dF_{b_i} = -\epsilon_{ijk} F_{B_j} \wedge dx^k \,, \quad dF_{B_i} = 0 \,, \quad dF_{c_i} = 0 \,, \quad dF_{\mathcal{C}_i} = -\epsilon_{ijk} F_{c_j} \wedge dx^k \,. \tag{231}$$

After gauging the 1-form symmetries, the coupling to the new gauge fields modifies the original action to the form

$$S = \int_X \left[ -\frac{1}{2e^2}(f_i - b_i) \wedge \star(f_i - b_i) - \frac{1}{2v^2}(\mathcal{F}_i - B_i) \wedge \star(\mathcal{F}_i - B_i) + c_i \wedge \star J_i^{\mathrm{m}} + \mathcal{C}_i \wedge \star \mathcal{J}_i^{\mathrm{m}} \right] \,. \tag{232}$$

Again, while this is fully gauge invariant under magnetic 1-form gauge transformations, invariance is violated by the electric 1-form transformation, which generates an 't Hooft anomaly,

$$\delta S = \frac{1}{2\pi} \int_X \left[ c \wedge (d\lambda_i^{\mathrm{e}} + \epsilon_{ijk} \Lambda_j^{\mathrm{e}} \wedge dx^k) + \mathcal{C}_i \wedge d\Lambda_i^{\mathrm{e}} \right] \,. \tag{233}$$

Gauge invariance can be restored by including the bulk SPT action

$$S_{\mathrm{bulk}} = -\frac{1}{2\pi} \int_Y \left[ dc_i \wedge b_i + (d\mathcal{C}^i - \epsilon_{ijk} c_j \wedge dx^k) \wedge B_i \right] \,, \tag{234}$$

where $\partial Y = X$, i.e. $X$ is the boundary of the five-dimensional manifold $Y$. Its 1-form gauge variation is

$$\delta S_{\mathrm{bulk}} = -\frac{1}{2\pi} \int_Y \left[ d\left( c_i \wedge \lambda_i^{\mathrm{e}} + \mathcal{C}_i \wedge d\Lambda_i^{\mathrm{e}} \right) + \epsilon_{ijk} \left( dc_i \wedge \Lambda_j^{\mathrm{e}} \wedge dx^k - c_j \wedge dx^k \wedge d\Lambda_i^{\mathrm{e}} \right) \right]$$

$$= -\frac{1}{2\pi} \int_Y d\left[ c_i \wedge (d\lambda_i^{\mathrm{e}} + \epsilon_{ijk} \Lambda_j^{\mathrm{e}} \wedge dx^k) + \mathcal{C}_i \wedge d\Lambda_i^{\mathrm{e}} \right] \,, \tag{235}$$

which precisely cancels (233) and restores gauge invariance of the full partition function. The corresponding anomaly polynomial is

$$I^{(6)} = \frac{1}{4\pi^2} \left( F_{c_i} \wedge F_{b_i} + F_{\mathcal{C}_i} \wedge F_{B_i} \right), \tag{236}$$

which is gauge invariant and closed, $dI^{(6)} = 0$, as follows from the fact that $2\pi I^{(6)} = d\mathcal{L}_{\text{bulk}}$.

## 4.5 Generalization to arbitrary spatial dimension

The above structure can be extended to general $d \geq 2$. In order to have a theory with a charge vector $q_i$, the fact that the $d$-dimensional Levi–Civita tensor $\epsilon$ has $d$ indices means we must extend the commutation relations (172) to the form

$$[iP_i, Q_{j_1 \dots j_r}] = \epsilon_{i j_1 \dots j_r k} q_k, \qquad [iP_i, q_j] = 0, \tag{237}$$

where to avoid clutter we have defined $r := d - 2$. To preserve the permutation symmetry of $\epsilon_{i_1 \dots i_d}$, the charge $Q_{i_1 \dots i_r}$ must be skew-symmetric in its $r$ indices. The commutation relations imply that a particle with a charge vector $\boldsymbol{q}$ cannot move in any direction except along the direction of $\boldsymbol{q}$, so again we expect such particles to be lineons.

The construction of the vector charge theory in $d$ spatial dimensions runs completely parallel to the $d = 3$ case, with the single index on the nonuniform charge (and associated quantities) replaced by $d - 2$ skew-symmetric indices. Following the previous discussion, we may similarly introduce current 1-forms $j_i$ and $K_{i_1 \dots i_r}$ for the charges $q_i$ and $Q_{i_1 \dots i_r}$, respectively, such that

$$q_i = \int_V \star j_i, \qquad Q_{i_1 \dots i_r} = \int_V \star K_{i_1 \dots i_r}, \tag{238}$$

where $V$ is now a $d$-cycle. Equation (237) implies that the currents are now related by

$$\star K_{i_1 \dots i_r} = \star J_{i_1 \dots i_r} - \epsilon_{j i_1 \dots i_r k} x_j \star j_k, \tag{239}$$

where $j_i$ and $J_{i_1 \dots i_r}$ are uniform and have conservation laws

$$d \star j_i = 0, \qquad d \star J_{i_1 \dots i_r} = \epsilon_{j i_1 \dots i_r k} dx_j \wedge \star j_k. \tag{240}$$

We can introduce a coupling to gauge fields via

$$S_{\text{cpl}} = \int_X \left( a_i \wedge \star j_i + \frac{1}{r!} \mathcal{A}_{i_1 \dots i_r} \wedge \star J_{i_1 \dots i_r} \right), \tag{241}$$

with gauge transformations

$$\delta a_i = d\sigma_i - \frac{1}{r!} \epsilon_{i j_1 \dots j_r k} \Sigma_{j_1 \dots j_r} dx_k, \qquad \delta \mathcal{A}_{i_1 \dots i_r} = d\Sigma_{i_1 \dots i_r}, \tag{242}$$

and gauge-invariant field strengths

$$f_i := da_i + \frac{1}{r!} \epsilon_{i j_1 \dots j_r k} \mathcal{A}_{j_1 \dots j_r} \wedge dx_k, \qquad \mathcal{F}_{i_1 \dots i_r} := d\mathcal{A}_{i_1 \dots i_r}, \tag{243}$$

respectively. An effective action is

$$S[a_i, \mathcal{A}_{i_1 \dots i_r}] = \int_X \left( -\frac{1}{2e^2} f_i \wedge \star f_i - \frac{1}{2r! v^2} \mathcal{F}_{i_1 \dots i_r} \wedge \star \mathcal{F}_{i_1 \dots i_r} \right), \tag{244}$$

and the resulting equations of motion and Bianchi identities are

$$-\frac{1}{e^2}d \star f_i = 0, \tag{245}$$

$$-\frac{1}{v^2}d \star \mathcal{F}_{i_1 \dots i_r} - \frac{1}{e^2}\epsilon_{ji_1 \dots i_r k}dx_j \wedge \star f_k = 0, \tag{246}$$

$$df_i - \frac{1}{r!}\epsilon_{ij_1 \dots j_r k}\mathcal{F}_{j_1 \dots j_r} \wedge dx_k = 0, \tag{247}$$

$$d\mathcal{F}_{i_1 \dots i_r} = 0. \tag{248}$$

In the low-energy limit, $\mathcal{A}_{i_1 \dots i_r}$ acquires a mass gap. Taking the $m^2 := v^2/e^2 \to \infty$ limit of (246), we find

$$\epsilon_{ji_1 \dots i_r k}dx_j \wedge \star f_k = 0, \tag{249}$$

which has components

$$\frac{1}{r!}\epsilon_{ij_1 \dots j_r k}(\mathcal{A}_{j_1 \dots j_r})_0 = (e_{[i})_{k]}, \tag{250}$$

$$\frac{1}{r!}\epsilon_{il_1 \dots l_r [j}(\mathcal{A}_{l_1 \dots l_r})_{k]} = \partial_{[j}(a_i)_{k]}. \tag{251}$$

As before, these can be inverted to obtain explicit expressions for the components of $\mathcal{A}_{i_1 \dots i_r}$. Contracting (251) with $\epsilon_{im_1 \dots m_r k}$, we find

$$\epsilon_{im_1 \dots m_r k}\partial_{[j}(a_i)_{k]} = \frac{1}{2r!}\epsilon_{im_1 \dots m_r k}\epsilon_{il_1 \dots l_r j}(\mathcal{A}_{l_1 \dots l_r})_k - \frac{1}{2r!}\epsilon_{im_1 \dots m_r k}\epsilon_{il_1 \dots l_r k}(\mathcal{A}_{l_1 \dots l_r})_j \tag{252}$$

$$= \frac{1}{2r!}\left[(r+1)!\delta_{m_1}^{[l_1} \dots \delta_{m_r}^{l_r}\delta_k^{j]}(\mathcal{A}_{l_1 \dots l_r})_k - 2!r!\delta_{m_1}^{[l_1} \dots \delta_{m_r}^{l_r]}(\mathcal{A}_{l_1 \dots l_r})_j\right] \tag{253}$$

$$= -\frac{1}{2}\left[(\mathcal{A}_{m_1 \dots m_r})_j + \sum_{n=1}^{r}\delta_{jm_n}(\mathcal{A}_{m_1 \dots i \dots m_r})_i\right], \tag{254}$$

where in the final sum, the $n$-th index $m_n$ in the Levi–Civita symbol is replaced by $i$. Note that the factors of $(r+1)!$ and $r!$ in the numerators of Eq. (253) are only present to compensate for the unit normalization of the skew-symmetrizations [...], where by definition we divide by the same factor. In other words, it simply means that there are $(r+1)!$ or $r!$ terms, each with relative prefactor 1.[25] The plus sign (overall minus sign) appears in front of the final sum in Eq. (254), since for each contribution in the first term of (253), we perform one swap of $l_n \leftrightarrow j$. Note that in writing the final term of Eq. (254), we have relabelled the dummy index $l_n$ as $i$. The remaining contribution from the first term of Eq. (253), in which $k$ is contracted with $j$, combines with $-2$ times itself from the second piece of Eq. (253) to give the first term of Eq. (254).

We now take a partial trace of Eq. (254) by contracting with $\delta_{m_r j}$. This gives

$$\epsilon_{im_1 \dots m_{r-1}jk}\partial_{[j}(a_i)_{k]} = -\frac{1}{2}\left[(\mathcal{A}_{m_1 \dots m_{r-1}j})_j + \sum_{n=1}^{r-1}\delta_{jm_n}(\mathcal{A}_{m_1 \dots i \dots m_{r-1}j})_i + d(\mathcal{A}_{m_1 \dots m_{r-1}i})_i\right]$$

$$= -2(\mathcal{A}_{m_1 \dots m_{r-1}j})_j. \tag{255}$$

Finally, we may insert Eq. (255) into Eq. (254) to solve for $(\mathcal{A}_{m_1 \dots m_r})_j$. Note that in Eq. (255), if we permute $j$ to any position among the indices $m_n$, any additional minus signs cancel as long as we perform the same permutation on both sides of the equation. Thus we can write

$$\sum_{n=1}^{r}\delta_{jm_n}(\mathcal{A}_{m_1 \dots l \dots m_r})_l = -\frac{1}{2}\sum_{n=1}^{r}\delta_{jm_n}\epsilon_{im_1 \dots l \dots m_r k}\partial_{[l}(a_i)_{k]}. \tag{256}$$

---

[25]Also recall that the factor 2! arises in the second term because we have summed over two indices, $i$ and $k$.

Evaluating the result, and similarly contracting Eq. (250) with $\epsilon_{im_1...m_r k}$, we obtain

$$(\mathcal{A}_{m_1...m_r})_0 = \frac{1}{2}\epsilon_{im_1...m_r k}(e_i)_k,$$

$$(\mathcal{A}_{m_1...m_r})_k = 2\left[\epsilon_{im_1...m_r j}\partial_{[j}(a_i)_{k]} - \frac{1}{4}\sum_{n=1}^{r}\delta_{m_n k}\epsilon_{im_1...l...m_r j}\partial_j(a_i)_l\right]. \tag{257}$$

Just as in the three-dimensional case, at low energies all components of $\mathcal{A}_{m_1...m_r}$ are given in terms of $a_i$. Note that we may also apply these results in $d = 2$ by setting $r = 0$.

We can also simplify some expressions by 'dualizing' the nonuniform charges, currents and 1-form potentials in their 'internal' spatial index labels, thus converting the $r$-form structure of the charges to that of a 2-form. Namely, we may contract the first commutation relation of Eq. (237) with $\epsilon_{lj_1...j_r m}$ to obtain[26]

$$[iP_i, \widetilde{Q}_{lm}] = \delta_{il}q_m - \delta_{im}q_l, \tag{259}$$

where we have defined

$$\widetilde{Q}_{lm} := \frac{1}{r!}\epsilon_{lj_1...j_r m}Q_{j_1...j_r}, \tag{260}$$

and we have chosen the order of indices such as to avoid unnecessary minus signs. Writing the charge as

$$\widetilde{Q}_{lm} = \int_V \star\widetilde{K}_{lm}, \tag{261}$$

the 'dual' nonuniform current can be expressed in terms of uniform currents via

$$\star\widetilde{K}_{lm} = \star\widetilde{J}_{lm} - x_l \star j_m + x_m \star j_l. \tag{262}$$

This implies the conservation equations

$$d\star j_i = 0, \qquad d\star\widetilde{J}_{lm} = dx_l \wedge \star j_m - dx_m \wedge \star j_l. \tag{263}$$

Again we may follow the same procedure as before. The dual currents can similarly be coupled to dual gauge fields via the action

$$S_{\text{cpl}} = \int_X \left(a_i \wedge \star j_i + \frac{1}{2}\widetilde{\mathcal{A}}_{lm} \wedge \star\widetilde{J}_{lm}\right), \tag{264}$$

which is invariant under the gauge transformations

$$\delta a_l = d\sigma_l - \widetilde{\Sigma}_{lm}dx_m, \qquad \delta\widetilde{\mathcal{A}}_{lm} = d\widetilde{\Sigma}_{lm}. \tag{265}$$

Comparing with Eq. (242), we see that the gauge parameters in the original and dual descriptions are related as

$$\widetilde{\Sigma}_{lm} = \frac{1}{r!}\epsilon_{li_1...i_r m}\Sigma_{i_1...i_r}, \tag{266}$$

which in either case corresponds to $d(d-1)/2$ components. Similarly, by comparing Eq. (243) with the gauge-invariant field strengths

$$f_l := da_l + \widetilde{\mathcal{A}}_{lm} \wedge dx_m, \qquad \widetilde{\mathcal{F}}_{lm} := d\widetilde{\mathcal{A}}_{lm}, \tag{267}$$

---

[26]Note that if we had $q_i = P_i$, this would reduce to the standard Euclidean algebra of rotations and translations,

$$[iP_i, \widetilde{Q}_{lm}] = \delta_{il}P_m - \delta_{im}P_l, \tag{258}$$

with $\widetilde{Q}_{lm}$ being identified as the angular momentum operators which generate rotations. A rotation is perhaps the most familiar example of a nonuniform symmetry.

we may read off the relations

$$\widetilde{\mathcal{A}}_{lm} = \frac{1}{r!}\epsilon_{li_1\ldots i_r m}\mathcal{A}_{i_1\ldots i_r}, \qquad \widetilde{\mathcal{F}}_{lm} = \frac{1}{r!}\epsilon_{li_1\ldots i_r m}\mathcal{F}_{i_1\ldots i_r}. \tag{268}$$

It is straightforward to check that the dual description yields equations of motion equivalent to (245) and (246), while the Bianchi identities (247) and (248) also follow.

In the dual description, the inverse Higgs constraint takes the simple form

$$(\widetilde{\mathcal{A}}_{ik})_0 = (e_{[i})_{k]}, \tag{269}$$

$$(\widetilde{\mathcal{A}}_{i[j})_{k]} = \partial_{[j}(a_i)_{k]}. \tag{270}$$

Despite appearances, the condition (270) actually fixes all components of $(\widetilde{\mathcal{A}}_{ij})_k$. To see this, consider the irreducible decomposition

$$(\widetilde{\mathcal{A}}_{ij})_k := H_{ijk} + S_{ijk}, \tag{271}$$

where $H_{ijk}$ is 'hook-symmetric', i.e. $H_{ijk} = -H_{jik}$ and $H_{[ijk]} = 0$, and $S_{ijk} = S_{[ijk]}$ is totally skew-symmetric. These symmetry properties imply that

$$H_{i[jk]} = -\frac{1}{2}H_{jki}, \qquad S_{i[jk]} = S_{ijk}. \tag{272}$$

Combining Eqs. (270), (271) and (272), we can solve for $(\widetilde{\mathcal{A}}_{ij})_k$ to obtain

$$(\widetilde{\mathcal{A}}_{ij})_k = \partial_j(a_{(k)}_i) - \partial_i(a_{(k)}_j) - \partial_k(a_{[i})_{j]}. \tag{273}$$

The light degrees of freedom are thus described by the hook-symmetric combination

$$\widetilde{A}_{ijk} := (\widetilde{\mathcal{A}}_{ij})_k + \partial_k(a_{[i})_{j]} = \partial_j a_{ik} - \partial_i a_{jk} = -(\widetilde{b}_k)_{ij}, \tag{274}$$

where $a_{ij} := (a_{(i)}_j)$ and

$$(\widetilde{b}_k)_{ij} := \frac{1}{r!}\epsilon_{im_1\ldots m_r j}(b_k)_{m_1\ldots m_r}, \tag{275}$$

is the Hodge dual of the magnetic field $b_k$ (199), generalized to $d$ spatial dimensions. Dualizing again, the corresponding quantity in the original theory is

$$A_{i_1\ldots i_r j} := (\mathcal{A}_{i_1\ldots i_r})_j + \frac{1}{2}\epsilon_{li_1\ldots i_r m}\partial_j(a_l)_m$$
$$= \epsilon_{li_1\ldots i_r m}\partial_m a_{lj} = (b_j)_{i_1\ldots i_r}. \tag{276}$$

Both (274) and (276) are invariant under $\Sigma$ ($\widetilde{\Sigma}$) gauge transformations, however they transform under shifts of $\sigma_i$ as

$$\delta\widetilde{A}_{ijk} = \partial_k\partial_{[j}\sigma_{i]}, \qquad \delta A_{i_1\ldots i_r j} = \frac{1}{2}\epsilon_{li_1\ldots i_r m}\partial_j\partial_m\sigma_l, \tag{277}$$

respectively. Similarly, we may write down an electric field,

$$(\widetilde{e}_i)_k := (\widetilde{\mathcal{A}}_{ik})_0 + \partial_0(a_{[i})_{k]} = \partial_k(a_i)_0 - \partial_0 a_{ik}, \tag{278}$$

whose gauge transformation depends on $\sigma_i$ only,

$$\delta(\widetilde{e}_i)_k = \partial_0\partial_{[k}\sigma_{i]}. \tag{279}$$

To summarize, we have constructed a generalization of the theory with a charge vector to any spatial dimension $d \geq 2$. The nonuniform charges have $d-2$ antisymmetric indices. This ensures immobility under every translation orthogonal to the charge vector, with the nonuniform charges filling out the other $d-2$ spatial directions, such that particles become lineons. It is clear that at low energies, spontaneous breaking of nonuniform 1-form symmetries will lead to a higher-dimensional analogue of the vector charge gauge theory, whose gapless degrees of freedom are provided by the symmetric part of the gauge fields $(a_{(i)}_j)$ associated to the charge vector.

## 4.6 Number of gapless modes

Let us count the number of gapless physical modes in this theory. For simplicity, let us first consider the case of $d = 3$. The number of spontaneously broken 1-form symmetries is $3 + 3 = 6$, and each of them produces $3 - 1 = 2$ modes. By the inverse Higgsing, the $3^2 = 9$ components of $(\mathcal{A}_i)_j$ are eliminated. Thus in total, there are $12 - 9 = 3$ gapless modes. In the vector charge gauge theory, the electric field is of rank-2 and symmetric, and there are $d(d + 1)/2$ components. Owing to Gauss' law, $d$ of them are eliminated. For $d = 3$, this leaves $6 - 3 = 3$ modes.

To extend this counting to arbitrary dimension, we should consider the more general theory with a $(d - 2)$-form (or equivalently a 2-form) of nonuniform charges (244). The number of such charges is given by

$$\begin{pmatrix} d \\ d - 2 \end{pmatrix} = \begin{pmatrix} d \\ 2 \end{pmatrix} = \frac{d(d - 1)}{2}. \tag{280}$$

Alongside the $d$ vector charges, each charge corresponds to a spontaneously broken 1-form symmetry and produces $(d - 1)$ gapless modes. However, at low energies the spatial components of $\mathcal{A}$ are gapped by the inverse Higgs mechanism. Thus, the total number of gapless modes remaining is

$$\overbrace{(d - 1) \times \left( d + \frac{d(d - 1)}{2} \right)}^{\text{1-form SSB}} \underbrace{- \frac{d^2(d - 1)}{2}}_{\text{Inverse Higgsing}} = \overbrace{\frac{d(d + 1)}{2}}^{\text{Sym. tensor of rank 2}} \underbrace{-d}_{\text{Gauss law}}, \tag{281}$$

agreeing with the number of gapless modes in the vector charge gauge theory generalized to $d$ dimensions.

## 5 Generalization to higher-pole gauge fields

Let us consider a generalization of the theory with $U(1)$ and dipole 1-form symmetries to higher-order moments. We will introduce generators of higher order moments, $Q_{i_1 \cdots i_n}$, where all the indices are symmetric, and refer to the symmetry generated by $Q_{i_1 \cdots i_n}$ as an "$n$-pole symmetry." In this terminology, a dipole symmetry is described as "1-pole" symmetry. We first discuss the case of $n = 3$, and then extend to generic values of $n$. In this section, we will explicitly write summation symbols.

### 5.1 Up-to-3-pole gauge theory

Here we consider a set of multipole charges up to 3-pole, $\{Q, Q_i, Q_{ij}, Q_{ijk}\}_{i,j,k=1,\cdots,d}$, where the indices are symmetric. The charges satisfy the following commutation relations with the generators of spatial translations:

$$[iP_i, Q_j] = \delta_{ij} Q, \tag{282}$$

$$[iP_k, Q_{ij}] = \delta_{ki} Q_j + \delta_{kj} Q_i, \tag{283}$$

$$[iP_l, Q_{ijk}] = \delta_{li} Q_{jk} + \delta_{lj} Q_{ki} + \delta_{lk} Q_{ij}. \tag{284}$$

To satisfy this algebra, the corresponding currents should take the form

$$\star K_i = \star J_i - x_i \star J, \tag{285}$$

$$\star K_{ij} = \star J_{ij} - x_i \star J_j - x_j \star J_i + x_i x_j \star J, \tag{286}$$

$$\star K_{ijk} = \star J_{ijk} - x_i \star J_{jk} - x_j \star J_{ki} - x_k \star J_{ij} + x_i x_j \star J_k + x_j x_k \star J_i + x_k x_i \star J_j - x_i x_j x_k \star J, \tag{287}$$

where the currents $\{J, J_i, J_{ij}, J_{ijk}\}_{i,j,k=1,\cdots,d}$ are uniform and the latter three are not conserved. Indeed, the currents above reproduce the algebra of charges, for example,

$$
\begin{aligned}
[iP_k, Q_{ij}] &= \left[iP_k, \int_V \star K_{ij}\right] \\
&= \delta_{ki}\int_V \star J_j + \delta_{kj}\int_V \star J_i - \int_V (\delta_{ki}x_j + \delta_{kj}x_i)\star J \\
&= \delta_{ki}\int_V \star K_j + \delta_{kj}\int_V \star K_i \\
&= \delta_{ki}Q_j + \delta_{kj}Q_i \,.
\end{aligned}
\tag{288}
$$

The conservation laws are written using the uniform currents as

$$
\mathrm{d}\star J = 0\,,
\tag{289}
$$

$$
\mathrm{d}\star J_i = \mathrm{d}x_i \wedge \star J\,,
\tag{290}
$$

$$
\mathrm{d}\star J_{ij} = \mathrm{d}x_i \wedge \star J_j + \mathrm{d}x_j \wedge \star J_i\,,
\tag{291}
$$

$$
\mathrm{d}\star J_{ijk} = \mathrm{d}x_i \wedge \star J_{jk} + \mathrm{d}x_j \wedge \star J_{ki} + \mathrm{d}x_k \wedge \star J_{ij}\,.
\tag{292}
$$

Let us introduce a coupling to gauge fields,

$$
\mathcal{L}_{\mathrm{cpl}} = \mathcal{A} \wedge \star J + \sum_i \mathcal{A}_i \wedge \star J_i + \sum_{i,j} \mathcal{A}_{ij} \wedge \star J_{ij} + \sum_{i,j,k} \mathcal{A}_{ijk} \wedge \star J_{ijk}\,,
\tag{293}
$$

where $\{\mathcal{A}, \mathcal{A}_i, \mathcal{A}_{ij}, \mathcal{A}_{ijk}\}_{i,j,k=1,\cdots,d}$ are 1-form gauge fields. Note that they are totally symmetric with respect to the indices $i, j, k \cdots$: $\mathcal{A}_{ij} = \mathcal{A}_{ji}$ and so on. To reproduce the conservation laws (289)–(292), the gauge transformations should be

$$
\delta\mathcal{A} = \mathrm{d}\Lambda + \sum_i \Lambda_i \mathrm{d}x^i\,,
\tag{294}
$$

$$
\delta\mathcal{A}_i = \mathrm{d}\Lambda_i + \sum_j 2\Lambda_{ij}\mathrm{d}x^j\,,
\tag{295}
$$

$$
\delta\mathcal{A}_{ij} = \mathrm{d}\Lambda_{ij} + \sum_k 3\Lambda_{ijk}\mathrm{d}x^k\,,
\tag{296}
$$

$$
\delta\mathcal{A}_{ijk} = \mathrm{d}\Lambda_{ijk}\,,
\tag{297}
$$

where the gauge parameters are symmetric with respect to the indices $i, j, k, \cdots$. Indeed, quadrupole gauge invariance implies that the quadrupole current is conserved,

$$
\begin{aligned}
\delta_{\Lambda_{ij}}\mathcal{L}_{\mathrm{cpl}} &= \sum_{i,j}\left(2\Lambda_{ij}\mathrm{d}x^j \wedge \star J_i + \mathrm{d}\Lambda_{ij} \wedge \star J_{ij}\right) \\
&= \sum_{i,j}\Lambda_{ij}\left(\mathrm{d}x^j \wedge \star J_i + \mathrm{d}x^i \wedge \star J_j - \mathrm{d}\star J_{ij}\right),
\end{aligned}
\tag{298}
$$

while octupole gauge invariance implies octupole current conservation,

$$
\begin{aligned}
\delta_{\Lambda_{ijk}}\mathcal{L}_{\mathrm{cpl}} &= \sum_{ijk}\left(3\Lambda_{ijk}\mathrm{d}x^k \wedge \star J_{ij} + \mathrm{d}\Lambda_{ijk} \wedge \star J_{ijk}\right) \\
&= \sum_{ijk}\Lambda_{ijk}(\mathrm{d}x^i \wedge \star J_{jk} + \mathrm{d}x^j \wedge \star J_{ki} + \mathrm{d}x^k \wedge \star J_{ij} - \mathrm{d}\star J_{ijk}).
\end{aligned}
\tag{299}
$$

The gauge-invariant field strengths are

$$\mathcal{F} = \mathrm{d}\mathcal{A} - \sum_i \mathcal{A}_i \wedge \mathrm{d}x^i \,, \tag{300}$$

$$\mathcal{F}_i = \mathrm{d}\mathcal{A}_i - \sum_j 2\mathcal{A}_{ij} \wedge \mathrm{d}x^j \,, \tag{301}$$

$$\mathcal{F}_{ij} = \mathrm{d}\mathcal{A}_{ij} - \sum_k 3\mathcal{A}_{ijk} \wedge \mathrm{d}x^k \,, \tag{302}$$

$$\mathcal{F}_{ijk} = \mathrm{d}\mathcal{A}_{ijk} \,. \tag{303}$$

Their Bianchi identities are given by

$$\mathrm{d}\mathcal{F} = -\sum_i \mathcal{F}_i \wedge \mathrm{d}x^i \,, \tag{304}$$

$$\mathrm{d}\mathcal{F}_i = -\sum_j 2\mathcal{F}_{ij} \wedge \mathrm{d}x^j \,, \tag{305}$$

$$\mathrm{d}\mathcal{F}_{ij} = -\sum_k 3\mathcal{F}_{ijk} \wedge \mathrm{d}x^k \,, \tag{306}$$

$$\mathrm{d}\mathcal{F}_{ijk} = 0 \,. \tag{307}$$

We can express the Bianchi identities as conservation laws,

$$\mathrm{d}(\mathcal{F} + \sum_i \mathcal{F}_i x_i + \sum_{i,j} \mathcal{F}_{ij} x_i x_j + \sum_{i,j,k} \mathcal{F}_{ijk} x_i x_j x_k) = 0 \,, \tag{308}$$

$$\mathrm{d}(\mathcal{F}_i + \sum_j 2\mathcal{F}_{ij} x_j + \sum_{j,k} 3\mathcal{F}_{ijk} x_j x_k) = 0 \,, \tag{309}$$

$$\mathrm{d}(\mathcal{F}_{ij} + \sum_k 3\mathcal{F}_{ijk} x^k) = 0 \,. \tag{310}$$

Let us consider a theory with dynamical $U(1)$ charge, dipole, quadrupole, and octupole gauge fields. We can write down an effective action,

$$S = \int_X \left( -\frac{1}{2(e_0)^2} \mathcal{F} \wedge \star \mathcal{F} - \frac{1}{2(e_1)^2} \sum_i \mathcal{F}_i \wedge \star \mathcal{F}_i - \frac{1}{2(e_2)^2} \sum_{i,j} \mathcal{F}_{ij} \wedge \star \mathcal{F}_{ij} \right.$$
$$\left. - \frac{1}{2(e_3)^2} \sum_{i,j,k} \mathcal{F}_{ijk} \wedge \star \mathcal{F}_{ijk} \right). \tag{311}$$

In this theory, we have the following Wilson line operators,

$$\mathrm{e}^{\mathrm{i}q \int_C \mathcal{A}} \,, \qquad \mathrm{e}^{\mathrm{i}\sum_i q_i \ell \int_C \mathcal{A}_i} \,, \qquad \mathrm{e}^{\mathrm{i}\sum_{ij} q_{ij} \ell^2 \int_C \mathcal{A}_{ij}} \,, \qquad \mathrm{e}^{\mathrm{i}\sum_{ijk} q_{ijk} \ell^3 \int_C \mathcal{A}_{ijk}} \,. \tag{312}$$

Except for the final one, they are not gauge-invariant unless $C$ lies purely in the time direction, i.e. they are fractons.

## 5.2 Up-to-$n$-pole gauge theory

Here we consider a gauge theory where $0, 1, \cdots, n$-pole symmetries are fully gauged. There are various possible choices of the algebra, depending on the mobility restrictions that one wants to implement. For example, in the theory discussed in Sec. 3.11, which reduces to the traceless scalar charge gauge theory, the quadrupole charges are partially gauged, in addition

to the charge and dipole symmetries. In the theory discussed in this section, all $m$-pole Wilson lines are fractonic except for $m = n$.

For notational convenience, let us introduce a symmetrization operator $\mathcal{S}_{i_1 \cdots i_n}$, which takes the symmetrized summation with respect to the indices $i_1 \cdots i_n$ in the expression if they are not yet symmetrized. If the indices are already symmetrized, $\mathcal{S}_{i_1 \cdots i_n}$ acts trivially. For example,

$$\mathcal{S}_{ij}[X_i Y_j] = X_i Y_j + X_j Y_i, \tag{313}$$

$$\mathcal{S}_{ij}[X_i X_j] = X_i X_j, \tag{314}$$

$$\mathcal{S}_{ijk}[X_i Y_j Z_k] = X_i Y_j Z_k + X_j Y_k Z_i + X_k Y_i Z_j + X_k Y_j Z_i + X_j Y_i Z_k + X_i Y_k Z_j, \tag{315}$$

$$\mathcal{S}_{ij}[X_i Y_{jk}] = X_i Y_{jk} + X_j Y_{ki} + X_k Y_{ij}, \tag{316}$$

where $Y_{ij} = Y_{ji}$.

We denote the $m$-pole charge by $Q_{i_1 \cdots i_m}$. The commutation relation of the translational generator and the $m$-pole charge is written as

$$\left[ i P_k, Q_{i_1 \cdots i_m} \right] = \mathcal{S}_{i_1 \cdots i_m} \left[ \delta_{k i_m} Q_{i_1 \cdots i_{m-1}} \right], \qquad \text{for } m \in \{1, \cdots, n\}, \tag{317}$$

and the 0-pole charge $Q$ commutes with $P_k$. To reproduce this algebra, the $m$-pole current should take the following form,

$$\star K_{i_1 \cdots i_m} = \mathcal{S}_{i_1 \cdots i_m} \left[ \sum_{\ell=0}^{m} (-)^\ell x_{i_1} \cdots x_{i_\ell} \star J_{i_{\ell+1} \cdots i_m} \right], \tag{318}$$

where $\star J_{i_1 \cdots i_m}$ are uniform currents. Here we use the convention $\star J_{i_m \cdots i_n} = \star J$ for $m > n$. The $m$-pole charge can be obtained by integrating $\star K_{i_1 \cdots i_m}$, and it indeed satisfies Eq. (317),

$$\begin{aligned}
\left[ i P_k, \int_V \star K_{i_1 \cdots i_m} \right] &= \mathcal{S}_{i_1 \cdots i_m} \left[ -\sum_{\ell=1}^{m} (-)^\ell \int_V \partial_k (x_{i_1} \cdots x_{i_\ell}) \star J_{i_{\ell+1} \cdots i_m} \right] \\
&= \mathcal{S}_{i_1 \cdots i_m} \left[ -\sum_{\ell=1}^{m} (-)^\ell \int_V \delta_{k i_1} x_{i_2} \cdots x_{i_\ell} \star J_{i_{\ell+1} \cdots i_m} \right] \\
&= \mathcal{S}_{i_1 \cdots i_m} \left[ \delta_{k i_m} \mathcal{S}_{i_1 \cdots i_{m-1}} \left[ \sum_{\ell=1}^{m} (-)^{\ell-1} \int_V x_{i_1} \cdots x_{i_{\ell-1}} \star J_{i_\ell \cdots i_{m-1}} \right] \right] \\
&= \mathcal{S}_{i_1 \cdots i_m} \left[ \delta_{k i_m} Q_{i_1 \cdots i_{m-1}} \right].
\end{aligned} \tag{319}$$

The conservation law of the current (318) can be written in the form

$$d \star J_{i_1 \cdots i_m} = \mathcal{S}_{i_1 \cdots i_m} \left[ dx_{i_1} \wedge \star J_{i_2 \cdots i_m} \right], \qquad \text{for } m \in \{1, \dots, n\}. \tag{320}$$

We show this by induction. For $m = 1$, the conservation law reads

$$d \star J_i = dx_i \wedge \star J, \tag{321}$$

and the expression (320) is true. Suppose that Eq. (320) holds up to $m$. The current conservation for $\star K_{i_1 \cdots i_{m+1}}$ reads

$$\begin{aligned}
d \star K_{i_1 \cdots i_{m+1}} &= \mathcal{S}_{i_1 \cdots i_{m+1}} \Big[ d \star J_{i_1 \cdots i_{m+1}} - dx_{i_1} \wedge \star J_{i_2 \cdots i_{m+1}} - x_{i_1} d \star J_{i_2 \cdots i_{m+1}} \\
&\qquad\qquad + dx_{i_1} x_{i_2} \star J_{i_3 \cdots i_{m+1}} + x_{i_1} x_{i_2} d \star J_{i_3 \cdots i_{m+1}} - \cdots \Big] \\
&= \mathcal{S}_{i_1 \cdots i_{m+1}} \Big[ d \star J_{i_1 \cdots i_{m+1}} - dx_{i_1} \wedge \star J_{i_2 \cdots i_{m+1}} - x_{i_1} dx_{i_2} \wedge \star J_{i_3 \cdots i_{m+1}} \\
&\qquad\qquad + dx_{i_1} x_{i_2} \star J_{i_3 \cdots i_{m+1}} + x_{i_1} x_{i_2} dx_{i_3} \wedge \star J_{i_4 \cdots i_{m+1}} \\
&\qquad\qquad - dx_{i_1} x_{i_2} x_{i_3} \star J_{i_4 \cdots i_{m+1}} + \cdots \Big] \\
&= 0,
\end{aligned} \tag{322}$$

where we used the expression (320) up to $m$. The colored parts (and the terms denoted by $\cdots$) in the second line cancel on symmetrization, and the conservation law is written as

$$\mathrm{d} \star J_{i_1 \cdots i_{m+1}} = \mathcal{S}_{i_1 \cdots i_{m+1}} \left[ \mathrm{d}x_{i_1} \wedge \star J_{i_2 \cdots i_{m+1}} \right] . \tag{323}$$

Thus, we have shown that Eq. (320) holds for every $m$.

We shall introduce a coupling of the currents $\{J_{i_1 \cdots i_m}\}_{m=1,\cdots,n}$ to 1-form gauge fields $\{\mathcal{A}_{i_1 \cdots i_m}\}_{m=1,\cdots,n}$ by

$$\mathcal{L}_{\mathrm{cpl}} = \sum_{m=0}^{n} \sum_{i_1,\cdots,i_m} \mathcal{A}_{i_1 \cdots i_m} \wedge \star J_{i_1 \cdots i_m} . \tag{324}$$

The gauge fields $\mathcal{A}_{i_1,\cdots,i_m}$ are symmetric under permutations of the indices. To reproduce the conservation law (320), the gauge fields should transform as

$$\delta \mathcal{A}_{i_1 \cdots i_m} = \mathrm{d}\Lambda_{i_1 \cdots i_m} + \sum_j (m+1)\Lambda_{i_1 \cdots i_m j} \mathrm{d}x^j . \tag{325}$$

The gauge-invariant field strengths are

$$\mathcal{F}_{i_1 \cdots i_m} = \mathrm{d}\mathcal{A}_{i_1 \cdots i_m} - \sum_j (m+1)\mathcal{A}_{i_1 \cdots i_m j} \wedge \mathrm{d}x^j . \tag{326}$$

With these gauge fields, we can construct the effective action,

$$S = -\int_X \left( \sum_{m=0}^{n} \sum_{i_1,\cdots,i_m} \frac{1}{(e_m)^2} \mathcal{F}_{i_1 \cdots i_m} \wedge \star \mathcal{F}_{i_1 \cdots i_m} \right) . \tag{327}$$

As a result of the gauge transformation (325), every $m$-pole Wilson line operator except for $m = n$,

$$\exp \left[ \mathrm{i} \sum_{i_1,\cdots,i_m} q_{i_1 \cdots i_m} \ell^m \int_C \mathcal{A}_{i_1 \cdots i_m} \right] , \tag{328}$$

is fractonic and can be placed only along the time direction.

## 5.3 Number of gapless modes

Here we count the number of physical gapless modes in the up-to-$n$-pole gauge theory (327). We show that this number coincides with the number of physical gapless modes in the rank-$(n+1)$ symmetric tensor gauge theory [14], which is consistent with the fact that the former theory reduces to the latter at low energies.

It is convenient to use a Young diagram to denote the number of independent components of a tensor representation corresponding to the diagram in $d$ spatial dimensions. For example,

$$\square = d , \quad \square\square = \frac{d(d+1)}{2} , \quad \begin{array}{c}\square\\\square\end{array} = \frac{d(d-1)}{2} , \quad \begin{array}{c}\square\square\\\square\end{array} = \frac{d(d+1)(d-1)}{3} , \quad \cdots . \tag{329}$$

As discussed in Sec. 3.4, in the case of up-to-dipole gauge theory, the number of gapless modes can be computed in two ways. We can represent Eq. (77) in Young-diagram notation as

$$\overbrace{(\square - 1) \otimes (1 + \square)}^{\text{1-form SSB}} - \underbrace{\begin{array}{c}\square\\\square\end{array}}_{\text{Inverse Higgs}} = \square\square - 1 . \tag{330}$$

This relation is equivalent to the following irreducible decomposition of tensor products,

$$\square \otimes (1 + \square + \square\square) = \square + \square\square + \square\square\square + \begin{array}{c}\square\\\square\end{array} + \begin{array}{c}\square\square\\\square\end{array}. \tag{331}$$

As we show blow, this structure extends to the up-to-$n$-pole gauge theory.

To discuss this, let us introduce a few notations. The number of independent components of a completely symmetric tensor of rank $n$ in $d$-spatial dimensions is given by

$$N(d,k) := \overbrace{\square\cdots\square}^{k} = \binom{d-1+k}{k}. \tag{332}$$

Let us consider the number of components of the tensor $\overbrace{\begin{array}{c}\square\square\cdots\square\\\square\end{array}}^{k}$. Let us pick $\ell \in \{1,\cdots,d-1\}$

and place it in the upper-left block, as $\overbrace{\begin{array}{c}\boxed{\ell}\,\square\cdots\square\\\square\end{array}}^{k}$. Then, we can place $\ell+1$ to $d$ in the box below

$\ell$, totaling $d-\ell$ candidates. To fill the horizontal row of boxes of length $k-1$ to the right of $\ell$,

$\overbrace{\square\square\cdots\square}^{k-1}$, we can use the numbers $\ell$ to $d$. Thus, for a given $\ell$, we have $(d-\ell)N(d-\ell+1,k-1)$
independent components. Summing them up, we obtain

$$
\begin{aligned}
M(d,k) &:= \overbrace{\begin{array}{c}\square\square\cdots\square\\\square\end{array}}^{k}\\
&= \sum_{\ell=1}^{d-1}(d-\ell)N(d-\ell+1,k-1)\\
&= \sum_{\ell=1}^{d-1}\ell\, N(\ell+1,k-1)\\
&= \sum_{\ell=1}^{d-1}\ell\binom{\ell+k-1}{k-1}\\
&= k\binom{d-1+k}{k+1}.
\end{aligned}
\tag{333}
$$

Here we have used

$$\sum_{\ell=1}^{d-1}\ell\binom{\ell+k-1}{k-1} = k\sum_{\ell=1}^{d-1}\binom{\ell+k-1}{k} = k\sum_{\ell=0}^{d-2}\binom{\ell+k}{k} = k\binom{d+k-1}{k+1}, \tag{334}$$

which follows from the so-called hockey-stick identity,

$$\sum_{\ell=0}^{n}\binom{\ell+k}{k} = \binom{n+k+1}{k+1}. \tag{335}$$

Let us count the number of gapless modes in the up-to-$n$-pole gauge theory. The SSB of a 1-form symmetry results in $d-1$ modes. The number of $k$-pole gauge fields, $\mathcal{A}_{i_1\cdots i_k}$, is given by $N(d,k)$. The inverse Higgs constraint for a $k$-pole gauge field is of the form $(\mathcal{A}_{i_1\cdots[i_k})_{j]} = (\text{Derivative of }(k-1)\text{-pole gauge field})$. The corresponding Young diagram for

$(\mathcal{A}_{i_1\cdots[i_k})_{j]}$ is of the form $\overbrace{\begin{array}{c}\square\square\cdots\square\\\square\end{array}}^{k}$ and the number of degrees of freedom is given by $M(d,k)$.

Alternatively, the up-to-$n$-pole gauge theory can be written using the completely symmetric electric field tensor of rank $n+1$, $E_{i_1\dots i_{n+1}}$. Each component is gauge invariant and produces one physical mode, up to one Gauss-law constraint, $\partial_{i_1}\cdots\partial_{i_{n+1}}E_{i_1\dots i_{n+1}}=0$. Thus, the number of modes can be counted as $N(d,n+1)-1$.

Since the two methods of counting are equivalent, we have the relation

$$(d-1)\sum_{k=0}^{n}N(d,k)-\sum_{k=1}^{n}M(d,k)=N(d,n+1)-1\,,\tag{336}$$

which is a direct generalization of the relation (330) for the up-to-dipole gauge theory. In Young-diagram notation, Eq. (336) is expressed as

$$\underbrace{(\,\square-1)\otimes(1+\square+\square\square+\cdots+\overbrace{\square\cdots\square}^{n})}_{\text{1-form SSB}}-\underbrace{\square-\square\square-\cdots-\overbrace{\square\square\cdots\square}^{n}}_{\text{Inverse Higgs constraints}}=\overbrace{\square\cdots\square}^{n+1}-1\,.\tag{337}$$

This is equivalent to the following irreducible decomposition,

$$\square\otimes(1+\square+\square\square+\cdots+\overbrace{\square\cdots\square}^{n})=\square+\square\square+\cdots+\overbrace{\square\cdots\square}^{n+1}+\square+\square\square+\cdots+\overbrace{\square\square\cdots\square}^{n}\,.\tag{338}$$

Equation (336) can be verified by a direct computation as follows:

$$\begin{aligned}\text{LHS}&=\sum_{k=0}^{n}\left((d-1)\binom{d-1+k}{k}-k\binom{d-1+k}{k+1}\right)\\&=\sum_{k=0}^{n}\left((d-1)\frac{(d-1+k)!}{k!(d-1)!}-k\frac{(d-1+k)!}{(k+1)!(d-2)!}\right)\\&=\sum_{k=0}^{n}\binom{d-1+k}{k+1}\\&=\sum_{k=0}^{n+1}\binom{d-2+k}{d-2}-1\\&=\binom{d+n}{n+1}-1\\&=N(d,n+1)-1\,,\end{aligned}\tag{339}$$

where we have used the identity (335).

# 6 Summary and discussions

In this paper, we have discussed a systematic method for realizing gapless fractonic phases through spontaneously broken nonuniform higher-form symmetries. These symmetries are characterized by the noncommutativity of their conserved charges with spatial translations and are a higher-form analogue of spacetime symmetries. The existence of fractons is rephrased as the existence of worldlines whose configurations are restricted. Such worldlines are the charged objects of 1-form symmetries. The mobility restrictions are fully controlled by the commutation relations of the corresponding charge operators with the generators of spatial translations. The resulting gauge theories have gapless excitations which play the role of

Nambu–Goldstone modes, some of which acquire a gap. This is a generalization of the inverse Higgs phenomenon known for spacetime symmetries to higher-form symmetries. Upon SSB of nonuniform higher-form symmetries, there appear emergent magnetic higher-form symmetries that are also nonuniform. In $(3 + 1)$ dimensions, these magnetic symmetries are also 1-form symmetries, and the corresponding magnetic monopoles receive mobility restrictions, which are controlled by the commutation algebra of the charges with translations. Similarly to the case of Maxwell theory, there is an 't Hooft anomaly between electric and magnetic symmetries, for which we have identified the corresponding bulk SPT action. The fractonic property is preserved under Higgsing. We have shown that by coupling the $U(1)$ and dipole gauge theory to charged matter and confining the dipole Wilson lines inside planes, the foliated field theory, which is an effective gauge theory for the X-cube model, can be derived.

The present formulation allows us to view existing models of gapless fractonic phases, such as scalar/vector charge gauge theories and their variants, from a unified perspective. These phases are understood within the symmetry-breaking paradigm, and mobility restrictions are controlled by the commutation relations of the corresponding charges with translations. We can also identify the underlying reason fractons appear in elasticity theory as being due to the emergent nonuniform magnetic 1-form symmetries associated with the spontaneously broken 0-form nonuniform symmetries. The method is systematic and allows us to engineer fractonic models with various desired mobility restrictions.

To conclude, we list several possible future directions.

- The method can be generalized to higher-dimensional fractonic objects straightfor­wardly. If we would like to restrict a line-like object, we may consider a 2-form sym­metry and restrict its motion by introducing other generators that do not commute with translations.

- It would be interesting to look for possible mixed 't Hooft anomalies of nonuniform symmetries with other kinds of symmetries, with which we would be able to obtain non­perturbative constraints on the possible IR phases for theories with this kind of symmetry.

- An important question is the stability of the broken phase at zero and finite tempera­tures. For the SSB of nonuniform higher-form symmetries, the dispersion relation of the Nambu–Goldstone modes is not always linear. For example, the gapless modes in the theory discussed in Sec. 4 obey the quadratic dispersion relation, $\omega \sim k^2$. The disper­sion relation is closely related to the stability of the broken phase. For the case of 0-form multipole symmetries, this issue has been discussed in Ref. [55].

- Although we have mainly considered gauge theories with continuous 1-form symmetries, the fractonic nature of a worldline remains after Higgsing, and the gauge fields therein can be used to construct a theory with fracton order, as we demonstrated in Sec. 3.10. It would be interesting to study to what extent those gauge fields can describe phases with fracton order.

- There have been studies on the formulation of hydrodynamic theories for fractonic sys­tems [56,57], and a theory of fracton magnetohydrodynamics (MHD) has recently been discussed [58]. For non-fractonic systems, an MHD can be formulated based on the con­servation laws of magnetic higher-form symmetries [59–62]. We have identified mag­netic higher-form symmetries in fractonic systems, and it would be interesting to formu­late a fractonic MHD based on them. For this purpose, it would be important to under­stand how these theories are coupled to curved spacetime, as discussed in Refs. [16,63] for dipole symmetries.

• It would be interesting to explore how the current formulation is related to other geometric constructions [64–66] as well as the associated interplay with gravitational physics.

## Acknowledgments

We thank Alfredo Pérez and Stefan Prohazka for useful comments.

**Funding information** Y. H. and S. A. are supported by an appointment of the JRG Program, and M. Y. is supported under the YST program, at the APCTP, which is funded through the Science and Technology Promotion Fund and Lottery Fund of the Korean Government, and is also supported by the Korean Local Governments of Gyeongsangbuk-do Province and Pohang City. Y. H. is also supported by the National Research Foundation (NRF) of Korea (Grant No. 2020R1F1A1076267) funded by the Korean Government (MSIT) and by JSPS KAKENHI Grant No. JP22H05111. S. A. also acknowledges support from the NRF of Korea (Grant No. 2022R1F1A1070999) funded by the Korean Government (MSIT). G.Y.C. is supported by the NRF of Korea (Grant No. 2020R1C1C1006048, No. RS-2023-00208291, No. 2023M3K5A1094810, No. 2023M3K5A1094813) funded by the Korean Government (MSIT) and by the Institute of Basic Science under project code IBS-R014-D1. G.Y.C. is also supported by the Air Force Office of Scientific Research under Award No. FA2386-20-1-4029 and No. FA2386-22-1-4061. G.Y.C also acknowledges Samsung Science and Technology Foundation under Project Number SSTF-BA2002-05.

## A  Coupling to complex scalar fields

The coupling procedure of the currents of nonuniform symmetries to gauge fields is applicable to generic matter field with the corresponding nonuniform symmetries. To facilitate comparison with literature, let us here discuss a concrete model with a dipole symmetry, the coupling of which to background gauge fields is discussed, e.g., in Ref. [67]. We here consider a model with a complex scalar field $\Phi$. The Lagrangian of the model is invariant under the transformation

$$\Phi \mapsto e^{i\Lambda}\Phi\,, \tag{A.1}$$

with $\lambda = \alpha + \beta_i x^i$, where $\alpha$ and $\beta_i$ are constants. To construct an invariant Lagrangian, let us consider the following combination,

$$\psi_{ij} := \Phi\partial_i\partial_j\Phi - \partial_i\Phi\partial_j\Phi\,. \tag{A.2}$$

Under a transformation with a local parameter $\Lambda(x)$,

$$\psi_{ij} \mapsto e^{2i\Lambda}(\psi_{ij} + \Phi^2 i\partial_i\partial_j\Lambda)\,. \tag{A.3}$$

We can construct dipole-invariant quantities such as $\psi_{ij}^*\psi_{ij}$ and $(\Phi^*)^2\psi_{ii}$. They are transformed under a transformation with local $\Lambda$ as

$$\begin{aligned}(\psi_{ij}^*\psi_{ij})' &= (\psi_{ij}^* - i\partial_i\partial_j\Lambda(\Phi^*)^2)(\psi_{ij} + i\partial_i\partial_j\Lambda\Phi^2) \\ &= \psi_{ij}^*\psi_{ij} + i\partial_i\partial_j\Lambda\left(\psi_{ij}^*\Phi^2 - \psi_{ij}(\Phi^*)^2\right) + \partial_i\partial_j\Lambda\partial_i\partial_j\Lambda|\Phi|^4\,,\end{aligned} \tag{A.4}$$

$$\left((\Phi^*)^2\psi_{ii}\right)' = (\Phi^*)^2\psi_{ii} + i\partial_i\partial_j\Lambda\,\delta_{ij}|\Phi|^4\,. \tag{A.5}$$

When the parameter is of the form $\Lambda = \alpha + \beta_i x^i$ with constant $\alpha$ and $\beta_i$, these quantities are indeed invariant. We can for example construct the following Lagrangian,

$$\mathcal{L} = |\dot\Phi|^2 - m^2|\Phi|^2 - \lambda\,\psi_{ij}^*\psi_{ij} - \lambda'((\Phi^*)^2\psi_{ii} + \Phi^2\psi_{ii}^*)\,. \tag{A.6}$$

The variation of the Lagrangian (A.6) with a local $\Lambda$ is

$$\begin{aligned}\delta\mathcal{L} &= i\left(\dot\Phi^*\Phi - \Phi^*\dot\Phi\right)\dot\Lambda - i\lambda\left(\psi_{ij}(\Phi^*)^2 - \psi_{ij}^*\Phi^2\right)\partial_i\partial_j\Lambda \\ &=: j^0\dot\Lambda - \tilde{J}^{ij}\partial_i\partial_j\Lambda\,.\end{aligned} \tag{A.7}$$

Here we have defined $\tilde{J}^{ij}$ by

$$\tilde{J}^{ij} = i\lambda\left(\psi_{ij}(\Phi^*)^2 - \psi_{ij}^*\Phi^2\right)\,. \tag{A.8}$$

Noting that $\delta\mathcal{L}$ can also be written as

$$\delta\mathcal{L} = (j^0\partial_t + j^i\partial_i)\Lambda\,, \tag{A.9}$$

we can see that the spatial components of the $U(1)$ current are given by

$$j^i = \partial_j\tilde{J}^{ij}\,. \tag{A.10}$$

To identify the dipole current, let us choose the transformation parameter to be $\Lambda(x) = \Sigma_i(x)x^i$, such that

$$\begin{aligned}\delta\mathcal{L} &= j^0 x^i\partial_t\Sigma_i + j^j\partial_j\left(\Sigma_i x^i\right) \\ &= \left(x^i j^0\partial_0 + x_i j^j\partial_j\right)\Sigma_i + j^i\Sigma_i\,.\end{aligned} \tag{A.11}$$

To ensure that the system has a dipole symmetry, when $\Sigma_i$ is a constant, $j^i\Sigma_i$ should be a total derivative, which we denote by $j^i = -\partial_\mu(k^i)^\mu$. We can now write the variation as

$$\delta\mathcal{L} = \left[(x^i j^0 + (k^i)^0)\partial_0 + (x_i j^j + (k^i)^j)\partial_j\right]\Sigma_i\,. \tag{A.12}$$

We can identify the conserved dipole current as

$$(K^i)^\mu = x^i j^\mu + (k^i)^\mu = \begin{pmatrix} x^i j^0 + (k^i)^0 \\ x^i j^j + (k^i)^j \end{pmatrix}\,. \tag{A.13}$$

The divergence of $(K^i)^\mu$ reads

$$\partial_\mu(K^i)^\mu = x^i\partial_\mu j^\mu + \partial_\mu(k^i)^\mu + j^i\,. \tag{A.14}$$

If we use the $U(1)$ current conservation, the dipole current conservation is written as

$$\partial_\mu(k^i)^\mu + j^i = 0\,, \tag{A.15}$$

which corresponds to Eq. (40).

Let us find the expression for $(k^i)^\mu$. The invariance under global dipole transformations implies that we can write the variation of the Lagrangian under a local $U(1)$ transformation as

$$\delta\mathcal{L} = j^0\partial_0\Lambda - \tilde{J}^{ij}\partial_i\partial_j\Lambda\,. \tag{A.16}$$

Now we take $\Lambda = \Sigma_i(x)x^i$. Noting that $\partial_i\partial_j(\Sigma_k x^k) = x_k\partial_i\partial_j\Sigma_k + \partial_i\Sigma_j + \partial_j\Sigma_i$,

$$\delta\mathcal{L} = \left[x^i j^0\partial_0 + \left(\partial_k\tilde{J}^{kj}x^i - \tilde{J}^{ij}\right)\partial_j\right]\Sigma_i - \partial_k\left(\tilde{J}^{kj}x^i\partial_j\Sigma_i\right)\,. \tag{A.17}$$

Noting that $j^i = \partial_j \tilde{J}^{ji}$, we can see that $(k^i)^0 = 0$ and $(k^i)^j = -\tilde{J}^{ij}$.

We can introduce the coupling to $U(1)$ and dipole gauge fields as

$$\mathcal{L}_{\text{cpl}} = a_\mu j^\mu + (\mathcal{A}^i)_\mu (k^i)^\mu. \tag{A.18}$$

The conservation of $U(1)$ and dipole currents can be enforced by imposing the invariance under the following gauge transformation,

$$\delta a_\mu = \partial_\mu \Lambda - \delta_{i\mu} \Sigma_i, \tag{A.19}$$

$$\delta(\mathcal{A}^i)_\mu = \partial_\mu \Sigma^i. \tag{A.20}$$

# B    Evaluation of order parameters

Here we provide additional detail on the evaluation of the order parameter in the theory discussed in Sec. 3, which has a $U(1)$ charge 1-form symmetry and a dipole 1-form symmetry.

## B.1    Low-energy Lagrangian and propagators

In order to compute the order parameters, we need the correlation functions of the gauge fields. We use the following Lagrangian density,

$$\mathcal{L} = \frac{1}{2(e_1)^2} (\partial_0 A_{ij} - \partial_i \partial_j a_0)^2 - \frac{1}{(e_1)^2} \partial^{[j} A_{ik]} \partial^{[j} A_{ik]}, \tag{B.1}$$

which can be obtained in the low energy limit of the theory with $U(1)$ and dipole gauge fields. We here change the normalization of $A_{ij}$ by introducing $A_{ij} = e_1 \widetilde{A}_{ij}$. Let us write the Lagrangian in the momentum space as

$$\mathcal{L} = \frac{1}{2} \begin{pmatrix} a_0(-\omega, -\boldsymbol{k}) & \widetilde{A}_{ij}(-\omega, -\boldsymbol{k}) \end{pmatrix} \frac{1}{(e_1)^2} \begin{pmatrix} k^4 & -ie_1 \omega k^k k^l \\ ie_1 \omega k^j k^i & (e_1)^2 \left[ \delta_{ik} \delta_{jl} (\omega^2 - k^2) + \delta_{il} k^k k^j \right] \end{pmatrix}$$
$$\times \begin{pmatrix} a_0(\omega, \boldsymbol{k}) \\ \widetilde{A}_{kl}(\omega, \boldsymbol{k}) \end{pmatrix}, \tag{B.2}$$

where $k := |\boldsymbol{k}|$. We define[27]

$$M := \frac{k^4}{(e_1)^2} \begin{pmatrix} 1 & -i\hat{e}_1 v \hat{k}^k \hat{k}^l \\ i\hat{e}_1 v \hat{k}^j \hat{k}^i & (\hat{e}_1)^2 \left[ \delta_{ik} \delta_{jl} (v^2 - 1) + \delta_{il} \hat{k}^k \hat{k}^j \right] \end{pmatrix}, \tag{B.3}$$

where $v := \omega/|\boldsymbol{k}|$, $\hat{k}_i := k_i/k$, and $\hat{e}_1 := e_1/k$. The following "longitudinal" vector is a zero mode of the matrix $M$,

$$\begin{pmatrix} iv\hat{e}_1 \\ \hat{k}_k \hat{k}_l \end{pmatrix}. \tag{B.4}$$

The transverse projection matrix is given by

$$P_T = \frac{1}{1 + (\hat{e}_1)^2 v^2} \begin{pmatrix} 1 & -iv\hat{e}_1 \hat{k}_k \hat{k}_l \\ iv\hat{e}_1 \hat{k}_i \hat{k}_j & (1 + (\hat{e}_1)^2 v^2) \delta_{ik} \delta_{jl} - \hat{k}_i \hat{k}_j \hat{k}_k \hat{k}_l \end{pmatrix}. \tag{B.5}$$

We try to find a matrix $N$ such that

$$NM = P_T. \tag{B.6}$$

---

[27]To simplify the notations, we do not explicitly indicate the symmetrization of components in the following. For example, in Eq. (B.3), the right-lower sector is implicitly made symmetric under $i \leftrightarrow j$, $k \leftrightarrow l$, and $ij \leftrightarrow kl$.

Let us parametrize $N$ as

$$N = \frac{(e_1)^2}{k^4} \begin{pmatrix} X & iZ_{ij} \\ -iZ_{ab} & Y_{abij} \end{pmatrix}, \tag{B.7}$$

where $Z_{ij}$ is symmetric, $Z_{ij} = Z_{ji}$, and $Y_{abij}$ is symmetric with respect to $a \leftrightarrow b$ and $i \leftrightarrow j$, and also $Y_{abij} = Y_{ijab}$. The following $X$, $Y_{abij}$, and $Z_{ab}$ satisfy Eq. (B.6),[28]

$$Y_{abij} = \frac{1}{(\hat{e}_1)^2(v^2 - 1)} \delta_{ai}\delta_{bj} + \frac{1}{(\hat{e}_1)^2 v^2(1 - v^2)} \delta_{ai}\hat{k}_b\hat{k}_j, \tag{B.8}$$

$$Z_{ab} = \frac{1}{\hat{e}_1 v(1 + (\hat{e}_1)^2 v^2)} \hat{k}_a\hat{k}_b, \tag{B.9}$$

$$X = \frac{2}{1 + (\hat{e}_1)^2 v^2}. \tag{B.10}$$

The two-point correlation function of $a_0$ is given by

$$\text{F.T.} \langle a_0(t, \boldsymbol{x})a_0(0, \boldsymbol{0}) \rangle = \frac{(e_1)^2}{k^4} \frac{2}{1 + (\hat{e}_1)^2 v^2} = \frac{2(e_1)^2}{k^4 + (e_1)^2 \omega^2}, \tag{B.11}$$

while the two-point function of $\widetilde{A}_{ab}(t, \boldsymbol{x})$ is written as

$$\begin{aligned}
\text{F.T.} \langle \widetilde{A}_{ab}(t, \boldsymbol{x})\widetilde{A}_{ij}(0, \boldsymbol{0}) \rangle &= \frac{m^2}{k^4} \left( \frac{1}{\hat{m}^2(v^2 - 1)} \delta_{ai}\delta_{bj} + \frac{1}{\hat{m}^2 v^2(1 - v^2)} \delta_{ai}\hat{k}_b\hat{k}_j \right) \\
&= \frac{1}{\omega^2 - k^2} \delta_{ai}\delta_{bj} + \left( \frac{1}{\omega^2} - \frac{1}{\omega^2 - k^2} \right) \delta_{ai}\hat{k}_b\hat{k}_j.
\end{aligned} \tag{B.12}$$

## B.2 Evaluation of the order parameters

Let us proceed with the evaluation of the correlation functions. We first look at the Wilson lines of fractons,

$$\lim_{\substack{T \to \infty \\ |\boldsymbol{x}| \to \infty}} \langle e^{i\int_{C_1} a} e^{-i\int_{C_2} a} \rangle \simeq \lim_{\substack{T \to \infty \\ |\boldsymbol{x}| \to \infty}} e^{-\frac{1}{2}\langle \left( \int_{C_1} a - \int_{C_2} a \right)^2 \rangle}. \tag{B.13}$$

We take the paths $C_1$ and $C_2$ to be straight lines along the time axis located at $\boldsymbol{x}$ and $\boldsymbol{0}$, respectively. The exponent of Eq. (B.13) is written as

$$\begin{aligned}
\frac{1}{2} \left\langle \left[ \int dt\, a_0(t, \boldsymbol{x}) - \int dt'\, a_0(t', \boldsymbol{0}) \right]^2 \right\rangle &= \int dt \int dt' \langle a_0(t, \boldsymbol{x})a_0(t', \boldsymbol{x}) \rangle \\
&\quad - \int dt \int dt' \langle a_0(t, \boldsymbol{x})a_0(t', \boldsymbol{0}) \rangle.
\end{aligned} \tag{B.14}$$

The Green function of $a_0$ in the IR is

$$\langle a_0(t, \boldsymbol{x})a_0(0, \boldsymbol{0}) \rangle = 2(e_1)^2 \int \frac{d\omega}{2\pi} \frac{d^d k}{(2\pi)^d} \frac{e^{-i\omega t + i\boldsymbol{k}\cdot\boldsymbol{x}}}{k^4 + (e_1)^2 \omega^2}. \tag{B.15}$$

In particular, we need its time integral,

$$\begin{aligned}
\int dt dt' \langle a_0(t, \boldsymbol{x})a_0(t', \boldsymbol{0}) \rangle &= 2(e_1)^2 \int dt dt' \int \frac{d\omega}{2\pi} \frac{d^d k}{(2\pi)^d} \frac{e^{-i\omega(t - t') + i\boldsymbol{k}\cdot\boldsymbol{x}}}{k^4 + (e_1)^2 \omega^2} \\
&= 2(e_1)^2 L_t \int \frac{d^d k}{(2\pi)^d} \frac{e^{i\boldsymbol{k}\cdot\boldsymbol{x}}}{k^4}.
\end{aligned} \tag{B.16}$$

---

[28]$Y_{abij}$ can contain a term $\hat{k}_a\hat{k}_b\hat{k}_k\hat{k}_j$ with arbitrary coefficient, which we take to be zero here.

The $|\boldsymbol{x}|$-dependence of the exponent is evaluated as

$$\frac{1}{2}\left\langle\left[\int dx^0 a_0(x^0,\boldsymbol{x})-\int dy^0 a_0(y^0,\boldsymbol{0})\right]^2\right\rangle = 2(e_1)^2 L_t \int \frac{d^d k}{(2\pi)^d}\frac{1-e^{i\boldsymbol{k}\cdot\boldsymbol{x}}}{\boldsymbol{k}^4}$$
$$\sim \begin{cases} |\boldsymbol{x}|^{4-d}, & d\neq 4, \\ \ln|\boldsymbol{x}|, & d=4, \end{cases} \quad \text{at large } |\boldsymbol{x}|. \tag{B.17}$$

Now let us turn to the evaluation of the order parameter for the dipole symmetry,

$$\left\langle e^{iq_i\ell\int_{C_1}\mathcal{A}_i}e^{-iq_j\ell\int_{C_2}\mathcal{A}_j}\right\rangle \simeq e^{-\frac{1}{2}\ell^2\left\langle\left(q_i\int_{C_1}\mathcal{A}_i-q_j\int_{C_2}\mathcal{A}_j\right)^2\right\rangle}. \tag{B.18}$$

We again take the same $C_1$ and $C_2$. We need the two-point correlation function of $(\mathcal{A}_i)_0$, which can be written at low energies as

$$\langle(\mathcal{A}_i)_0(t,\boldsymbol{x})(\mathcal{A}_j)_0(0,\boldsymbol{0})\rangle = \langle\partial_i a_0(t,\boldsymbol{x})\partial_j a_0(0,\boldsymbol{0})\rangle$$
$$= 2(e_1)^2 \int \frac{d\omega}{2\pi}\frac{d^d k}{(2\pi)^d}\frac{k_i k_j e^{-i\omega t+i\boldsymbol{k}\cdot\boldsymbol{x}}}{\boldsymbol{k}^4+(e_1)^2\omega^2}. \tag{B.19}$$

Using this, we have

$$\ell^2 q_i q_j\langle\int dt\int dt'(\mathcal{A}_i)_0(t,\boldsymbol{x})(\mathcal{A}_j)_0(t',\boldsymbol{0})\rangle = 2(e_1)^2\ell^2 L_t q_i q_j \int \frac{d^d k}{(2\pi)^d}\frac{k_i k_j e^{i\boldsymbol{k}\cdot\boldsymbol{x}}}{\boldsymbol{k}^4}$$
$$= 2(e_1)^2\ell^2 L_t q_i q_j \frac{1}{d}\delta_{ij}\int \frac{d^d k}{(2\pi)^d}\frac{e^{i\boldsymbol{k}\cdot\boldsymbol{x}}}{\boldsymbol{k}^2}. \tag{B.20}$$

The $|\boldsymbol{x}|$-dependence of the exponent of Eq. (B.18) is evaluated as

$$\frac{\ell^2}{2}\left\langle\left(q_i\int_{C_1}\mathcal{A}_i-q_j\int_{C_2}\mathcal{A}_j\right)^2\right\rangle = \frac{2}{d}(e_1)^2\ell^2 L_t \boldsymbol{q}^2 \int \frac{d^d k}{(2\pi)^d}\frac{1-e^{i\boldsymbol{k}\cdot\boldsymbol{x}}}{\boldsymbol{k}^2}$$
$$\sim \begin{cases} |\boldsymbol{x}|^{2-d}, & d\neq 2, \\ \ln|\boldsymbol{x}|, & d=2, \end{cases} \quad \text{at large } |\boldsymbol{x}|. \tag{B.21}$$

## C Transformation property of dipole Wilson lines

Here we detail the computation of the transformation property (122) of the dipole Wilson line operator (117). The symmetry generators of the $U(1)$ charge and dipole 1-form symmetries are written as

$$Q(S) := -\frac{1}{e^2}\int_S dS^i \widetilde{e}_i, \qquad Q_i(S) := -\frac{1}{(e_1)^2}\int_S dS^j (\mathcal{E}_i)_j - \frac{1}{e^2}\int_S dS^j x_i\widetilde{e}_j, \tag{C.1}$$

where $\widetilde{e}_i = -f_{0i} = f^{0i}$ is the electric field for the field strength $f = da + \mathcal{A}_i\wedge dx^i$, and $(\mathcal{E}_i)_j := \partial_j(\mathcal{A}_i)_0 - \partial_0(\mathcal{A}_i)_j$ is the dipole electric field. They satisfy the following canonical commutation relations,

$$\left[\frac{1}{e^2}\widetilde{e}_i(t,\boldsymbol{x}),a_j(t,\boldsymbol{y})\right] = i\delta_{ij}\delta(\boldsymbol{x}-\boldsymbol{y}), \qquad \left[\frac{1}{(e_1)^2}(\mathcal{E}_i)_j(t,\boldsymbol{x}),(\mathcal{A}_k)_l(t,\boldsymbol{y})\right] = i\delta_{ik}\delta_{jl}\delta(\boldsymbol{x}-\boldsymbol{y}). \tag{C.2}$$

We here show that

$$[iQ_i(S), \int_C \left( \partial_j a - \mathrm{d}a_j + (\mathcal{A}_l)_j \mathrm{d}y^l \right)] = \delta_{ij} I(S,C),$$

(C.3)

where $S$ is a 2-cycle and $C$ is a 1-cycle. In the following, we denote the coordinates for $S$ and $C$ by $\boldsymbol{x}$ and $\boldsymbol{y}$, respectively. The contribution from each term can be computed as follows.

- The first term gives

$$\begin{aligned}
\left[ iQ_i(S), \int_C \mathrm{d}y^k \partial_j a_k \right] &= \left[ -\frac{i}{e^2} \int_S \mathrm{d}S^l x_i \tilde{e}_l, \int_C \partial_j a_k \mathrm{d}y^k \right] \\
&= \int_S \mathrm{d}S^l \int_C \mathrm{d}y^k x_i \delta_{lk} \partial_j^{(y)} \delta(\boldsymbol{x}-\boldsymbol{y}) \\
&= \delta_{ij} \int_S \mathrm{d}S^l \int_C \mathrm{d}y^l \delta(\boldsymbol{x}-\boldsymbol{y}) \\
&= \delta_{ij} I(S,C),
\end{aligned}$$

(C.4)

  where $\partial_j^{(y)}$ denotes the derivative with respect to the coordinate $\boldsymbol{y}$. Here we used $\partial_j^{(y)} \delta(\boldsymbol{x}-\boldsymbol{y}) = -\partial_j^{(x)} \delta(\boldsymbol{x}-\boldsymbol{y})$ and also performed a partial integration.

- The second term can be computed in a similar way as

$$\begin{aligned}
\left[ iQ_i(S), -\int_C \mathrm{d}y^k \partial_k a_j \right] &= \left[ -\frac{i}{e^2} \int_S \mathrm{d}S^l x_i \tilde{e}_l, -\int_C \mathrm{d}y^k \partial_k a_j \right] \\
&= -\int_S \mathrm{d}S^l \int_C \mathrm{d}y^k x_i \delta_{lj} \partial_k^{(y)} \delta(\boldsymbol{x}-\boldsymbol{y}) \\
&= -\int_S \mathrm{d}S^l \int_C \mathrm{d}y^k \delta_{ik} \delta_{lj} \delta(\boldsymbol{x}-\boldsymbol{y}) \\
&= -\int_S \mathrm{d}S^j \int_C \mathrm{d}y^i \delta(\boldsymbol{x}-\boldsymbol{y}).
\end{aligned}$$

(C.5)

- The third term gives

$$\begin{aligned}
\left[ iQ_i(S), \int_C (\mathcal{A}_l)_j \mathrm{d}y^l \right] &= \left[ -\frac{i}{(e_1)^2} \int_S \mathrm{d}S^k (\mathcal{E}_i)_k, \int_C (\mathcal{A}_l)_j \mathrm{d}y^l \right] \\
&= \int_S \mathrm{d}S^k \int_C \mathrm{d}y^l \delta_{il} \delta_{kj} \delta(\boldsymbol{x}-\boldsymbol{y}) \\
&= \int_S \mathrm{d}S^j \int_C \mathrm{d}y^i \delta(\boldsymbol{x}-\boldsymbol{y}).
\end{aligned}$$

(C.6)

The contributions from the second and third terms cancel, and we obtain Eq. (C.3), which implies (122).

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
