# Peer review of "A symmetry principle for gauge theories with fractons"

_SciPost Physics, doi:SciPost Phys. 16, 050 (2024)_

## Round 2 · Referee Report · Anonymous (Referee 2) · 2022-9-26

Strengths

  1. The work provides a systematic treatment of gauge theories for non-uniform symmetries (e.g. dipole symmetries,) and their extended operators as worldlines for particles with restricted mobility.

  2. The detailed discussion of the gapless fractons as a result of the spontaneous symmetry breaking of the non-uniform higher form symmetries is very illuminating .

Report

Overall the paper is well written, presents a new comprehensive and unifying treatment of non-uniform gauge theories, and meets all the criteria for publication.

I do however have several comments detailed bellow which I think the authors should address before publication.

Requested changes

  1. The use of the same symbol $Q$ to denote both the charge of the original global symmetry and the 1-form symmetry resulting from gauging is very confusing. Not only are these genuinely different symmetries in different theories that act on different operators, but they may even have different symmetry groups. I highly recommend distinguishing between the two charges in some way, perhaps by calling the electric 1-form symmetry charges $Q^e$, similar to how the magnetic 1-form symmetry charges are denoted by $Q^m$.

  2. The discussion regarding the normalization of the dipole gauge field (on page 10) seems overly simplistic, and it is not clear to me that the normalization condition of $a$ is independent of the dipole gauge transformations. While the given computation showing why this is the case for the sphere is accurate (eq. 48), it seems like the same argument would fail on a torus that wraps around a non-trivial cycle (as $\int_{S_1} n_i dx^i \neq 0$ for the non-trivial cycle.) In general one would expect that the dipole symmetry could not be $U(1)$ just from the dipole transformations, and this squares with recent discussions of the various continuum limits of a dipole symmetry in [2201.10589]. It would be useful if the authors clarified why they can take the dipole symmetry to be $U(1)$ rather than $\mathbb{R}$.

  3. The authors don't discuss these gauge theories in the presence of charged matter apart from the Higgs phase. In particular the coupling of dipole invariant matter to tensor gauge fields was systematically constructed in [1807.11479], but it is not clear how the additional gauge fields of the non-uniform dipole gauge theory fit into this construction, or if there is a universal gauge principle for coupling these gauge theories to charged matter. I think it would be useful to discuss how to gauge the dipole symmetry of simple dipole invariant scalar theories in the spirit of [1807.11479], and present the resulting theory of matter coupled to the complete dipole gauge theory.

  4. On page 6 the authors state that "$q_z$ has non-vanishing commutation relations with translations," which directly contradicts equation (20). I think the statement they are trying to make is that $q_z$ is the commutator of a different conserved charge with translations, which results in particles charged under $q_z$ to be immobile in certain directions. Similar imprecise statements appear in the vector charge gauge theory section.

  5. The authors discuss t' Hooft anomalies of the 1-form symmetries. However there is also a possibility of t' Hooft (or mixed) anomalies between the 0-form symmetries that obstructs the gauging of the non-uniform symmetry in the first place, which the authors do not discuss at all. It would be interesting if the authors had anything to say about the possibility of such anomalies appearing, and what their consequences may be.

  • validity: -
  • significance: -
  • originality: -
  • clarity: -
  • formatting: -
  • grammar: -

Author:  Yuji Hirono  on 2023-11-28  [id 4155]

(in reply to Report 2 on 2022-09-26)

Please find our response to the comments in the attached file.

Attachment:

response2.pdf

---

## Round 2 · Referee Report · Anonymous (Referee 1) · 2022-9-26

Strengths

Interesting application of higher form symmetries to gapless fractonic theories

Weaknesses

Some technical aspects are not sufficiently spelled out, or unclear

Report

The authors consider higher-form symmetries that do not commute with translations, which they dub as ''nonuniform higher-form symmetries''. More concretely, they start with nonuniform 0-form symmetries and their 1-form conserved currents, which have an explicit dependence on the coordinates. Then they introduce a set of Abelian gauge fields coupled to the currents and postulate gauge transformations that are consistent with the conservation equations of the currents. Promoting the gauge fields to be dynamical, they construct an invariant action for them using the field strengths and derive the equations of motion and Bianchi identities. Using this last set of equations they identify electric and magnetic 1-form symmetries acting on Wilson or 't Hooft line operators. They identify the gapless modes of the dynamical gauge fields as the Nambu-Goldstone bosons of the spontaneously broken 1-form symmetries, analogously to electromagnetism.

From the invariance of Wilson (and 't Hooft) lines under (magnetic dual) gauge transformations they deduce the mobility constraints on (magnetically) charged particles, which agree with those proposed before for fractons in theories with nonuniform symmetries. They also derive 't Hooft anomalies of 1-form symmetries by introducing external gauge fields in the partition function.

They study in detail cases with a scalar and dipole Abelian charges and with vector charges in 3+1 dimensions, and introduce generalizations to theories containing higher multipole charges and in other dimensions. They connect with other descriptions of gapless fractonic phases based on higher-rank tensor fields in a low energy limit, and associate the necessary reduction in the number of degrees of freedom to an inverse Higgs mechanism originating in the explicit coordinate dependence of the charges.

The paper introduces an interesting point of view on the description of gapless phases with fractons, in particular in the identification of 1-form symmetries and magnetic charges. There are however some technical aspects that would benefit from some clarification before the paper is considered for publication.

Requested changes

1) I do not understand the first equality in (8). Suppose the space is $\mathbb{R}_t\times M^3$, and $j$ is a one-form. Denoting with $i$ the $M^3$ directions, and taking $\sqrt{g_3}$ the square root of the metric determinant, then, naively,

$$ \int_{M^3} {\cal L}{e^\mu} (\star j)=\int}d^3x {\cal L{e^\mu} (\sqrt{g_3}j^t)=\intj^\alpha)\right) $$}d^3x \left(e^\mu \partial_\mu (\sqrt{g_3}j^t)-\nabla_\alpha e^\mu (\sqrt{g_3
If $e^\mu=\delta_i^\mu$ in general $\nabla_\alpha e^\mu=\delta_i^\mu \Gamma_{\alpha i}^i \neq 0$ if $M^3$ is curved. I have a similar issue with the equality in (111).

2) This is maybe a typo, in (56) one is taking variations with respecto to $da$ and $dA$, but the formulas suggests the variations are respect to $a$ and $A$.

3) I'm not sure I agree with the comment below (50) relative to changing coefficients of time and space derivatives, it is not clear how one would maintain gauge invariance, at least with all the formulas that are provided in the language of differential forms.

4) I'm not sure about the conditions in (60) and (179) and how are they derived from the equations of motion (52)-(55) and (175)-(178), for instance, imposing $f=0$ in (54) does not give an additional constraint? Same for (177) when $f_i=0$.

5) Can the authors show that the equations of motion obtained from the Lagrangians (67) and (194) coincide with the equations (52)-(55) and (175)-(178)? Given the gauge fixings and change of variables involved it is not immediately obvious.

  • validity: ok
  • significance: good
  • originality: good
  • clarity: good
  • formatting: good
  • grammar: good

Author:  Yuji Hirono  on 2023-11-28  [id 4154]

(in reply to Report 1 on 2022-09-26)

Please find our response to the comments in the attached file.

Attachment:

response1.pdf

---

## Round 3 · Referee Report · Anonymous (Referee 1) · 2023-12-18

Report

The authors have answered in detail the points raised in the previous report. They have corrected some minor issues that do not affect to their results and have extended their analysis to include additional matter fields. In my opinion the paper can be published without further changes.

---

## Round 3 · Author Response

Dear Editor,

We thank the reviewers for the careful reading of the manuscript and the helpful feedback. In response to the reviewers' comments, we have updated out manuscript. Our responses to both reviewers can be found in the reply to the reports.

Sincerely,
Yuji Hirono, Minyoung You, Stephen Angus, and Gil Young Cho

---

## Round 3 · List of Changes

• We added a new appendix titled "Coupling to complex scalar fields" as Appendix A in which we discussed the coupling of gauge fields of non-uniform symmetries to a theory with a complex scalar field.

  • We added a new footnote on page 4 in which we comment on the case of curved spacetime.

  • We have fixed a typo in Eq. (56) of the previous manuscript (Eq. (57) in the updated one).

  • We have added a new comment in Summary and Discussions (the second point in the list of possible future directions on page 37).

  • We have replaced the phrasing "has non-vanishing commutation relations with translations," with "can be written as a commutator of a translation and another charge" on page 6 and page 22.

  • We have Footnote 3 on page 6 with the following content: "We will use the same symbol $Q$ to denote the charge of a 0-form symmetry and the charge of the corresponding 1-form symmetry that appears as a result of the gauging of the former, to emphasize the connection between these two symmetries. When we wish to highlight the degree of the symmetry, we explicitly write the dependence on the underlying manifold over which the charge density is integrated, e.g. $Q(V)$ and $Q(S)$, where $V$ and $S$ are a $d$-cycle and a $(d-1)$-cycle, respectively."

  • We have corrected Eq. (191), and we have modified the discussion prior to Eq. (191). We have also added footnote 21 following Eq. (192) to clarify our conventions.

  • We have replaced Eq. (249), fixed Eqs. (250) and (269), and modified the surrounding arguments.

  • We added a derivation of the equations of motion of the scalar charge gauge theory from the equations of motion (53)-(56) in page 12 (see around Eq. (69)-(76)).

---

## Editorial Decision

published